# DDI-Gaussian: Distributed Dynamic Instance Gaussian for Autonomous Driving

## Abstract

Current dynamic 3D Gaussian approaches attempt to decompose scene motion by using purely implicit features or additional 3D annotations. However, these strategies hinder fine-grained and interpretable control over Gaussians and overlook the inherent instance consistency, spatial continuity, and local rigidity in scene motion. To this end, we propose a novel framework that achieves MLP-free Gaussian instance distinction and effectively disentangles dynamic urban street scenes. Specifically, we assign each Gaussian a compact multi-hot instance feature, enabling direct differentiation without relying on an additional network. To model transient motions, we initialize sparse control points at the instance level and construct the motion field from coarse to fine by leveraging spatiotemporal relationships. Additionally, we introduce two instance-level losses: an instance-level region loss and an instance-semantic loss. The former, combined with the opacity rendering pipeline, enables precise instance-level rendering and suppresses ghosting artifacts. The latter enforces cross-view feature consistency and optimizes the spatial positions of instances. Notably, our framework avoids costly 3D instance annotations by instead utilizing 2D pseudo-labels generated by Segment Anything Model (SAM) for supervision. Our demo video and code are available in the anonymous repository at `https://anonymous.4open.science/r/DDI-GS-750E/`.

## 1 Introduction

Understanding open-world 3D scenes from monocular video has significant implications for robotics, virtual reality, gaming, filmmaking, and autonomous driving (Li et al., 2022; Wei et al., 2023; Tian et al., 2025). Neural Radiance Fields (NeRF) (Mildenhall et al., 2021) implicitly learn scene geometry and appearance from multi-view images, achieving high-quality reconstructions by predicting color and opacity at numerous sampled points along a ray. As an alternative, 3D Gaussian Splatting (3DGS) (Kerbl et al., 2023) models scenes using explicit 3D Gaussian primitives, delivering superior reconstruction fidelity along with efficient training and rendering performance. However, both approaches encounter substantial challenges in 3D scene reconstruction when applied to large-scale, dynamic autonomous driving scenarios.

To reconstruct dynamic scenes, many methods (Li et al., 2025b; Yang et al., 2024; Li et al., 2024; Fischer et al., 2024) introduce a deformable network to compute attribute offsets. However, when the model first constructs the static scene as an anchor, dynamic objects produce ghosting artifacts along their motion paths. Consequently, in the early stage of dynamic scene training, the model tends to take a shortcut by preserving these ghosting artifacts, which significantly hinders its ability to learn accurate motion patterns. The motion ghosting artifacts are illustrated in Figure.1. Some approaches (Turki et al., 2023; Yan et al., 2023) learn motion through scene decomposition. These works utilize tracked bounding boxes to separate static and dynamic objects, yet the reliance on expensive annotations limits their applicability. Some approaches (Guo et al., 2024; Feng et al., 2025) extend the 3D Gaussian mathematical model to 4D representations, but their rendering quality remains unsatisfactory. Methods such as Zhang et al. (2024) and Wu et al. (2024) factorize the 4D spatio-temporal tensor (XYZT) into six Hexplanes, achieving improved reconstruction quality. However, the aforementioned approaches overlook the instance-level and locally consistent nature of motion. This fine-grained understanding of the scene not only supports downstream applications, but also facilitates more accurate scene reconstruction. To enable instance-level control of

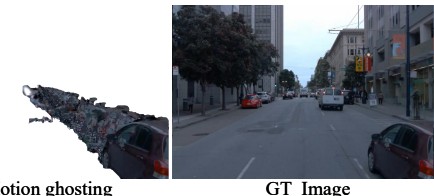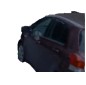

Reconstructed Image      Motion ghosting      GT Image      Ours

Figure 1: Qualitative visualization of dynamic ghosting in instance-level rendering. When modeling dynamic objects, the model tends to generate ghosting artifacts of the object at different positions. Our method effectively addresses this issue at the instance level. GT denotes the ground truth.

Gaussians, gaussian-grouping (Ye et al., 2024) assigns each Gaussian a high-dimensional instance feature, which is then processed through multi-layer perceptrons (MLPs) to obtain its instance ID. However, applying these to millions of Gaussians in autonomous driving scenarios results in an prohibitive computational burden. Overall, current approaches to reconstructing dynamic scenes encounter three principal challenges: (1) In dynamic scenes, existing methods tend to produce ghosting artifacts along motion trajectories, resulting in residual appearances of objects in future positions. Rather than capturing true motion patterns, these artifacts are merely pseudo-dynamic representations. (2) Lack of fine-grained scene and motion decomposition. Traditional methods overlook instance-level attributes that remain consistent across viewpoints, which is crucial for both dynamic scene reconstruction and downstream applications. (3) Memory and computational bottlenecks. In dynamic scenes for autonomous driving, millions of Gaussians require network-based computation of attribute offsets or instance IDs, resulting in a substantial computational burden.

To address these challenges, we propose decomposing both scene structure and motion at the instance level, inspired by three fundamental properties of real-world motion: instance coherence, spatial continuity, and local rigidity. However, a key question arises: **how can we effectively model and simplify motion at the instance level while enabling fast and accurate discrimination between Gaussians?**

Specifically, we propose an Distributed Dynamic Instance Gaussian (DDI-Gaussian), as illustrated in Figure 2. The proposed method jointly optimizes instance attributes using 2D instance ID masks and instance-segmented RGB images. We first introduce a learnable multi-hot encoding for instance representation, reformulating the high-dimensional feature classification task as a low-dimensional label-fitting problem. This enables direct differentiation between Gaussians in 3D space while circumventing reliance on the MLP. We further leverage the instance-segmented RGB images together with an instance-semantic loss to optimize the spatial distribution of instances and feature consistency. Building on this, we decompose dynamic scenes and suppress ghosting artifacts. To simplify motion, we perform sparse sampling to generate instance-level control points, which are then combined with an MLP to model scene motion. A dense motion field is subsequently interpolated from the spatial relationships between Gaussians and control points. To improve rendering quality and eliminate ghosting artifacts, we introduce an instance-level region loss and an opacity map rendering pipeline for accurate instance-level rendering. The entire framework is implemented based on the Grendel (Zhao et al., 2025) distributed framework. Notably, due to the high cost and sparsity of manual 3D annotations, we leverage 2D masks automatically generated by the Segment Anything Model (SAM) (Kirillov et al., 2023) as supervision.

In general, our contributions are as follows:

- We propose Distributed Dynamic Instance Gaussian, which not only enables instance-level perception but also efficiently reconstructs high-fidelity urban street scenes.

- We propose a novel Gaussian instance representation that enables fast discrimination of Gaussians without relying on MLPs, and captures transient motions effectively in a coarse-to-fine manner through instance-level control points.

- We introduce an instance-level region loss combined with an opacity rendering pipeline to enable precise instance-level rendering. Additionally, an instance-semantic loss is incorporated to regularize by leveraging feature consistency.

- Comprehensive experiments are conducted on challenging datasets including NoTR and Waymo, demonstrating the state-of-the-art performance of our method and the effectiveness of each component.

## 2 RELATED WORKS

### 2.1 3D GAUSSIAN SPLATTING

3DGS models a scene as millions of Gaussian primitives, achieving superior novel view synthesis quality along with fast training and rendering. Grendel-GS (Zhao et al., 2025) enables Gaussians rendering in a distributed manner. TCLC-GS (Zhao et al., 2024) fully utilizes LiDAR and camera sensors to enable high-quality 3D reconstruction. Through the integration of SAM and identity encoding, Gaussian Grouping (Ye et al., 2024) supports high-quality reconstruction, segmentation, and object editing. However, these approaches struggle to reconstruct dynamic scenes.

Current dynamic 3DGS research focuses primarily on deformation-based networks (Chu et al., 2024; Lu et al., 2024), 4D Gaussians (Guo et al., 2024; Duan et al., 2024), and flow-based methods (Zhu et al., 2024; Wang et al., 2025; Gao et al., 2024). Deformable 3D Gaussians (Yang et al., 2024) divide a dynamic scene into a static 3DGS and a deformable network, capturing temporal changes by predicting time-varying attributes for each Gaussian. GFlow (Gao et al., 2024) extends 3D to an explicit 4D representation by integrating depth and optical flow, enabling dynamic scene modeling. 4D Gaussian Splatting discretizes time into separate intervals, modeling each timestamp independently to extend 3D Gaussian primitives into 4D. However, these methods face challenges in training and rendering speed. To alleviate efficiency issues, SC-GS (Huang et al., 2024b) and Superpoint-GS (Wan et al., 2024) represent multiple deformable 3D Gaussians with a shared control point, thereby significantly reducing the computational overhead. Although SC-GS and Superpoint-GS significantly reduce computational overhead by representing similarly deformed 3D Gaussians with shared control points, they are difficult to scale to large scenes. Grid4D (Xu et al., 2024) mitigate computational costs through grid-based feature representations, but they lack fine-grained control over scene content and are limited by single-GPU constraints.

Therefore, we propose a distributed, instance-level framework driven by sparse control points to achieve accurate and efficient reconstruction of large-scale dynamic scenes.

### 2.2 SIMULATION ENVIRONMENTS FOR AUTONOMOUS DRIVING

Existing autonomous driving simulation engines, such as CARLA (Dosovitskiy et al., 2017) and AirSim (Shah et al., 2017), face challenges including the high manual labor cost of creating virtual environments and a lack of realism in the generated data. Novel view synthesis techniques like 3DGS and NeRF offer promising solutions. Mega-NeRF (Turki et al., 2022) and CityGaussian (Liu et al., 2024) adopt a divide-and-conquer strategy by segmenting scenes into separate blocks. StreetGaussian (Yan et al., 2023) models foreground and background separately and employs 4D spherical harmonics to alter vehicle appearances across frames. $S^3$Gaussian (Huang et al., 2024a) encodes spatiotemporal features using space-only and space-time planes. However, these methods lack fine-grained rendering. Moreover, in large-scale dynamic scenes, applying MLPs to all Gaussians not only slows down rendering, but also places higher demands on GPU memory.

## 3 METHODOLOGY

### 3.1 PRELIMINARIES

3DGS represents 3D scene information using colored Gaussians, demonstrating powerful scene reconstruction capabilities. However, its potential in scene understanding and dynamic scenarios remains to be fully explored.

Our research shows that instance-level dynamic 3DGS holds great promise for open-world, dynamically complex scenes. To represent the scene, each Gaussian is characterized by a 3D center position $\mu$ and a 3D covariance matrix. The covariance matrix is decomposed into components for optimization: $R$ serves as the rotation matrix, and $S$ serves as the scale matrix. Each Gaussian is assigned

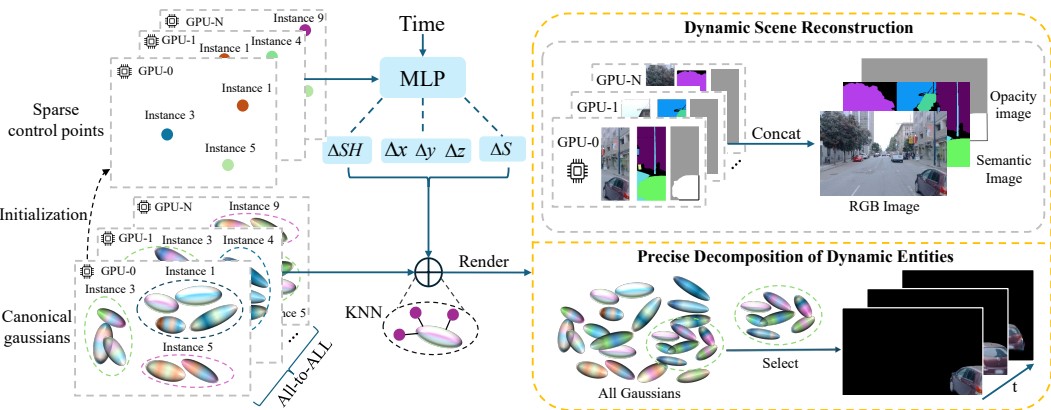

Figure 2: Overview of Distributed Dynamic Instance Gaussian. Each Gaussian and control point is assigned an instance attribute to enable instance-level association. Then, a dense motion field is constructed based on instance-level sparse control points and their spatial relationships. The entire process is supervised with pseudo-labels generated by SAM (Kirillov et al., 2023) and is implemented based on the Grendel (Zhao et al., 2025) distributed framework.

an opacity $\alpha$, view-independent semantic $e_i$ and instance features $o_i$, and view-dependent spherical harmonics. Thus, the scene can be parameterized as a set of Gaussians $\{G_i = \mu_i, q_i, s_i, \alpha_i, c_i, e_i, o_i\}$. It implements differentiable rendering and gradient-based optimization for each pixel in a sequence of Gaussians sorted in the depth order, for computing the final color and opacity.

It is worth noting that we use semantic features $e_i$ and instance features $o_i$ to separately learn the instance-segmented RGB images and instance IDs. Here, the semantic features $e_i$ corresponds to the DC component of the spherical harmonic coefficients representing the instance segmentation colors.

## 3.2 DISTRIBUTED DYNAMIC INSTANCE 3D GAUSSIAN

**Instance Encoding with Direct Distinguishability**. To generate spatio-temporally consistent instance masks, we introduce a learnable instance embedding for each Gaussian primitive. Unlike Ye et al. (2024), our method employs a compact multi-hot vector of length $K$, which serves as the instance representation. This design transforms the task of classifying high-dimensional features into a low-dimensional label fitting problem, enabling not only more compact and flexible encoding but also direct discrimination of Gaussian primitives without relying on MLPs. Specifically, the instance embedding is activated by a Sigmoid function. During training, continuous instance logits are optimized to fit the multi-hot instance encoding labels. When selecting Gaussians, the instance embedding is binarized channel-wise to obtain the instance IDs, enabling effective and explicit object distinction within the scene. Compared to approaches that predict a fixed number of instance features, the proposed method offers superior scalability. Following Kerbl et al. (2023), the instance encoding of each pixel is calculated by $N$ ordered points using $\alpha$-blending:

$$O_{id} = \sum_{i \in \mathcal{N}} o_i \alpha_i' \prod_{j=1}^{i-1} (1 - \alpha_j'), \tag{1}$$

where $O_{id}$ denotes the rendered 2D instance mask, $o_i$ is the instance encoding of the $i$-th Gaussian, and $\alpha'$ is the final opacity, obtained by multiplying the 2D covariance $\Sigma^{2D}$ with opacity $\alpha_i$. The $\Sigma^{2D}$ is derived from the 3D covariance matrix $\Sigma^{3D}$ through the following projection:

$$\Sigma^{2D} = JW\Sigma^{3D}W^T J^T, \tag{2}$$

where $J$ is the Jacobian of the affine approximation of the 3D-2D projection, and $W$ is the world-to-camera transformation matrix.

To reduce reliance on expensive 3D annotations, we use 2D masks generated by SAM as the ground truth. Leveraging SAM's powerful zero-shot capabilities, the proposed approach avoids the limitations of pre-defined training categories and allows greater flexibility . To ensure the uniqueness

of instance IDs, following Gaussian grouping, we use a well-trained zero-shot tracker Cheng et al. (2023) to associate instance masks across different views.

**Deformation network with sparse instance control points**. In autonomous driving scenarios, object motion patterns vary significantly in complexity across different instances, while often exhibiting locally consistent variations. Motivated by this, we propose to model motion at the instance level as the fundamental unit. Specifically, building upon static scene modeling, we first partition Gaussians into distinct instances using instance encoding. Then, for each instance, we apply the Farthest Point Sampling (FPS) algorithm to sample sparse control points according to the scaling factor $\beta$, allowing flexible adjustment and enabling a more compact and efficient representation. Notably, these control points inherit the same attribute features as the original Gaussian primitives. To model motion dynamics, we feed both the timestamp and control point positions into a lightweight MLP that predicts attribute offsets, thereby constructing a sparse motion field.

Based on the sparse motion field, we employ the Linear Blend Skinning (LBS) (Sumner et al., 2007) method to derive an dense motion field. For a $G_i$ belonging to the $i$-th instance, we first identify its $K$ nearest control points within the corresponding control point set using the k-nearest neighbors (KNN) algorithm. Interpolation weights are then computed based on the relative distances to these control points, and the final offset variation is obtained through weighted interpolation. To reduce computational cost, we restrict the KNN ($K = 3$) search to control points within the same instance. At time $t$, this process can be mathematically formalized as:

$$\Delta F_i^t = \sum_{k=1}^{K} \frac{\exp(-||\mu_i - c_{ik}||)}{\sum_{j=1}^{K} \exp(-||\mu_i - c_{ij}||)} \cdot \Delta f_k^t, \tag{3}$$

where $\Delta F_i^t$ and $\mu_i$ respectively represent the attribute offset and center position of the $i$-th Gaussian at time $t$, while $c_i$ and $\Delta f_i$ denote the center position and attribute offset of the corresponding control points for the $i$-th Gaussian.

**Distributed Training and Rendering**. We implement a dynamic distributed system based on the Zhao et al. (2025) and PyTorch's distribueted framework to balance computation across canonical Gaussians, control points, and the deformable network.

Specifically, canonical Gaussians and control points are evenly distributed across GPUs. During dynamic scene reconstruction, all control points from each GPU are aggregated to form a global set. Each Gaussian then identifies its $K$ nearest control points within the same instance to compute attribute offsets. Each GPU independently updates the Gaussians assigned to it while rendering a distinct image region. As the viewpoint changes, the $i$-th GPU obtains the necessary Gaussians residing on other GPUs via all-to-all communication. During training, the per-pixel average computation time on each GPU is calculated by dividing the rendering time by the number of pixels in its assigned region. This metric is then used to dynamically redistribute pixel regions, ensuring a balanced computational load. More details and experiments can be found in the Appendix.

Additionally, for fair comparison, we constrain the number of Gaussians for both the proposed method and the baselines within a comparable range, ensuring that performance gains are not simply due to using more Gaussians.

**Difference with Gaussian Grouping**. It is important to note that while both our method and Gaussian Grouping assign instance attributes, they differ in two fundamental aspects: (1) **Motivation and implement**. Our method is driven by the need for instance-level scene understanding, which is essential for accurate reconstruction of dynamic scenes in autonomous driving scenarios. To this end, we adopt a compact multi-hot vector for instance encoding and reformulate the feature classification task as a label prediction problem. This design removes the reliance on MLPs, allowing efficient and scalable instance differentiation through vectorized operations. Additionally, we initialize control points based on instance features to better handle large-scale environments. In contrast, Gaussian Grouping relies on an MLP to predict each instance ID from its features, which constrains scalability in large dynamic scenes. (2) **Loss function**. For dynamic driving scenes, we design an instance-level region loss to achieve precise rendering and suppress ghosting artifacts caused by moving objects. Additionally, we introduce a semantic-instance loss to leverage the correlation between instance and semantic features. In contrast, Gaussian Grouping focuses on the static spatial distribution of Gaussians constrains instance features through static spatial distributions. And, the loss is inaccurate in object boundary regions.

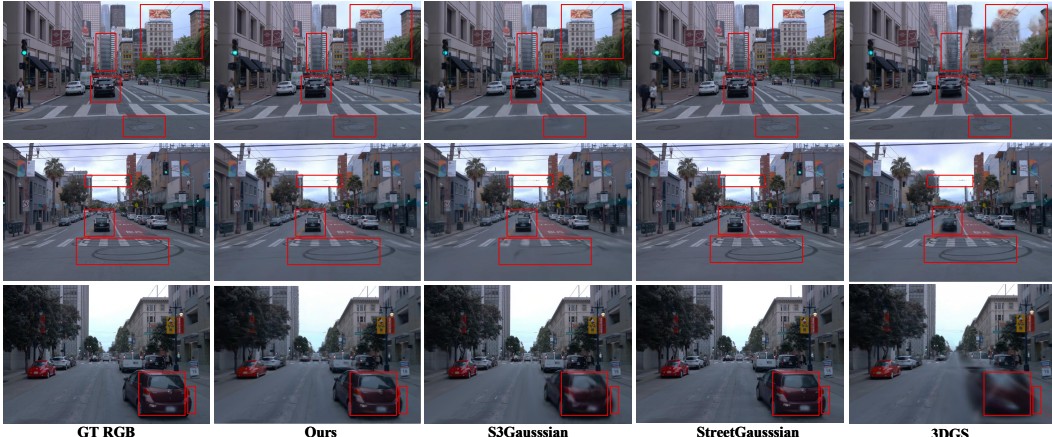

GT RGB      Ours      S3Gausssian      StreetGausssian      3DGS

Figure 3: Qualitative results. 3DGS fails to reconstruct dynamic objects effectively. S$^3$Gaussian and StreetGaussian produce blurry results in certain regions, with StreetGaussian additionally exhibiting ghosting artifacts along object boundaries. In contrast, our proposed method achieves high fidelity and sharp detail in the rendered results.

### 3.3 OPTIMIZATION

To stabilize training, we first build a coarse static scene, followed by the initialization and fine-tuning of control points. During the fine-tuning, an instance-level region loss is applied to achieve accurate rendering and suppress ghosting. Finally, we construct the dynamic scene. In addition to the image loss $\mathcal{L}_{color}$ defined in Kerbl et al. (2023), we optimize instance features using BCE loss to enhance the scene understanding capability of 3DGS. To better reconstruct dynamic scenes, we further propose an instance-semantic loss and an instance-level region loss.

**Instance-semantic Consistency Loss.** Since Gaussians belonging to the same instance are expected to share similar semantic features, this strong consistency can be used as a mutual constraint. Specifically, we compute the semantic centroid for each instance by aggregating the semantic features of its Gaussians, and then constrain the distance between each Gaussian's semantic feature and its corresponding instance centroid. This unsupervised approach enforces semantic consistency within the same instance, reduces internal semantic fragmentation, and improves segmentation coherence. We formalize the instance-sematic loss with $m$ sampling points as:

$$\mathcal{L}_{is} = \frac{1}{m} \sum_{i=1}^{m} ||e_i - \frac{1}{P} \sum_{p \in \mathcal{P}} e_p||, \tag{4}$$

where $e$ denotes the semantic feature and $\mathcal{P}$ represents the set of Gaussians (excluding the $i$-th one) among the $m$ sampled points that belong to the same instance as the $i$-th Gaussian, with $P$ being the number of elements in $\mathcal{P}$.

**Instance-Level Region Loss.** During dynamic scene optimization, a common issue is the emergence of false motion ghosting artifacts. For moving objects, many static Gaussians may remain along the object's motion trajectory. As a result, the object exhibit multiple motion states over time, overlapping in the rendered view as the camera moves. However, these Gaussians remain static residuals rather than exhibiting true motion. This makes the model outputs prone to approaching zero, thereby hindering the learning of motion patterns and even leading to model collapse.

To address this, we introduce an instance-level region loss. Specifically, we render Gaussians belonging to the same instance ID independently, and apply 2D instance masks $\mathcal{G}$ to penalize ghosting artifacts that do not correspond to the object's appearance at the time $t$. This loss, combined with independent rendering, ensures that object motion is caused solely by the predictions of the deformable network, rather than by static residuals observed at different time steps as the camera moves, thereby suppressing static residuals and enhancing motion fidelity. Specifically, we render Gaussians associated with the same instance ID independently and apply 2D instance masks $\mathcal{G}$ to penalize ghosting artifacts that do not correspond to the object's appearance at time $t$. Formally, the

Table 1: SOTA comparison on the Waymo-NOTR dataset. "PSNR*" and "SSIM*" denote the PSNR and SSIM of dynamic objects respectively. St-GS and S3-GS denote StreetGaussian and S$^3$Gaussian, respectively. "-" indicates that data cannot be obtained.

| Data | Metrics | Scene Reconstruction | | | | | Novel View Synthesis | | | | |
|------|---------|------|------|------|------|------|------|------|------|------|------|
| | | 3DGS | MARS | St-GS | S3-GS | Ours | 3DGS | MARS | St-GS | S3-GS | Ours |
| D32 | PSNR ↑ | 28.47 | 28.24 | 29.17 | 31.35 | **32.74** | 25.14 | 26.61 | 26.98 | 27.44 | **28.19** |
| | SSIM ↑ | 0.876 | 0.866 | 0.873 | 0.911 | **0.926** | 0.813 | 0.796 | 0.838 | 0.857 | **0.884** |
| | LPIPS ↓ | 0.136 | 0.252 | 0.138 | 0.106 | **0.095** | 0.165 | 0.305 | 0.149 | 0.137 | **0.115** |
| | PSNR* ↑ | 23.26 | 23.37 | 27.78 | 26.02 | 28.12 | 20.48 | 22.21 | **24.62** | 22.92 | 23.70 |
| | SSIM* ↑ | 0.716 | 0.701 | **0.818** | 0.783 | 0.811 | 0.655 | 0.697 | **0.742** | 0.680 | 0.702 |
| S32 | PSNR ↑ | 29.42 | 28.31 | - | 30.73 | **31.06** | 26.82 | **27.63** | - | 27.05 | 27.42 |
| | SSIM ↑ | 0.891 | 0.879 | - | 0.883 | **0.894** | 0.836 | **0.848** | - | 0.825 | 0.841 |
| | LPIPS ↓ | 0.118 | 0.196 | - | 0.116 | **0.113** | 0.134 | 0.193 | - | 0.142 | **0.132** |

instance-level region loss is as follows:

$$\mathcal{L}_{ir} = \frac{\sum_{i,j} tanh(\lambda \cdot f_{i,j}) \cdot (1 - \mathbb{I}(\mathcal{G}^t = \text{ID}))}{\sum_{i,j}(1 - \mathbb{I}(\mathcal{G}^t = \text{ID}))}, \tag{5}$$

where $\mathbb{I}$ denotes the indicator function, $\lambda$ is a hyperparameter, and $f_{i,j}$ represents the rendered result at pixel location $(i,j)$. In the proposed approach, $f$ corresponds to the opacity $\alpha$, and an additional opacity map rendering pipeline is introduced. The objective is to minimize opacity in non-target regions, thereby encouraging the removal of redundant Gaussians.

In the opacity map pipeline, the gradient with respect to $\alpha'$ is given by $T_i(2\alpha_i - \sum_{j=i+1}^{N} \alpha_j \alpha'_j \prod_{l=i+1}^{j-1}(1 - \alpha'_l))$. The derivation details are shown in Appendix A.4. Notably, this incurs almost no additional computational cost.

## 4 EXPERIMENTS

### 4.1 EXPERIMENTAL SETUP

**Datasets and metrics**. To evaluate our method in driving scenarios that better reflect real-world complexity and object motion, we conduct experiments on the NOTR (Yang et al., 2023) subset of the Waymo Open Dataset (Sun et al., 2020). The NOTR's dynamic32 (D32) and static32 (S32) datasets include a variety of challenging driving conditions such as ego-static scenes, high-speed motion, exposure mismatch, and rainy weather. Compared to datasets like nuScenes (Caesar et al., 2020) and nuPlan (Caesar et al., 2021), NOTR contains a greater number of dynamic objects. We evaluated our method on Waymo-Street (Yan et al., 2023) with different camera setups and resolutions. Details are provided in the Appendix. To assess the quality of the scenes, we synthesize novel views adjacent to the input frames and compute Peak Signal-to-Noise Ratio (PSNR), Structural Similarity (SSIM) (Wang et al., 2004), and Learned Perceptual Image Patch Similarity (LPIPS) (Zhang et al., 2018). To validate the generalization ability, we conducted additional real-world experiments using UAVs, with dataset descriptions and experimental details provided in Appendix A.2.

**Implementation details**. Our implementation follows the setup described in Huang et al. (2024a); Yan et al. (2023). We first construct a static scene to initialize the control points. During training, we employ the Adam optimizer (Kingma, 2014) for both Gaussian and deformable network optimization, with learning rates set to $1.6 \times 10^{-3}$ and $1.6 \times 10^{-4}$, respectively. The instance and semantic encoding have lengths of 8 and 3, respectively. The control point sampling ratio $\beta$ is set to 0.01. We conduct all experiments using two NVIDIA GeForce RTX 4090 GPUs, except for FPS evaluation.

**Baseline methods**. We compare our method with several state-of-the-art (SOTA) approaches. MARS(Wu et al., 2023) is a NeRF-based method that constructs dynamic scenes using 2D bounding boxes. OmniRe (Chen et al., 2024) and StreetGaussian (Yan et al., 2023) explicitly model dynamic scenes with bounding boxes. S$^3$-Gauassian (Huang et al., 2024a) and 4D-GS (Wu et al., 2024) employs HexPlane to decompose spatiotemporal 4D variables. 3DGS (Kerbl et al., 2023) and

Table 2: Quantitative comparison of scene reconstruction results on the Waymo dataset. $D_{opt}$ denote depth optimization. Box indicates methods that utilize bounding boxes for dynamic modeling.

| Methods | Venue | Box | $D_{opt}$ | Scene Reconstruction | | | Novel View Synthesis | | |
|---|---|---|---|---|---|---|---|---|---|
| | | | | PSNR↑ | SSIM↑ | LPIPS↓ | PSNR↑ | SSIM↑ | LPIPS↓ |
| 4D-GS | CVPR2024 | ✗ | ✗ | 31.01 | 0.890 | 0.205 | 28.16 | 0.870 | 0.219 |
| Deform-GS | CVPR2024 | ✗ | ✗ | 32.66 | 0.912 | 0.158 | 30.19 | 0.894 | 0.167 |
| Octree-GS | TPAMI2025 | ✗ | ✗ | 33.63 | 0.936 | 0.103 | 32.13 | **0.927** | 0.107 |
| Desire-GS | CVPR2025 | ✓ | ✓ | 34.55 | 0.926 | 0.108 | 32.56 | 0.919 | 0.119 |
| St-GS | ECCV2024 | ✓ | ✗ | 35.66 | 0.936 | 0.109 | 30.87 | 0.882 | 0.139 |
| S3-GS | ICCV2025 | ✗ | ✓ | 33.44 | 0.927 | 0.118 | 31.62 | 0.917 | 0.124 |
| OmniRe | ICLR2025 | ✓ | ✓ | 34.95 | **0.943** | 0.114 | 31.79 | 0.905 | 0.125 |
| Ours | N/A | ✗ | ✗ | **36.12** | 0.942 | **0.075** | **34.49** | 0.927 | **0.082** |

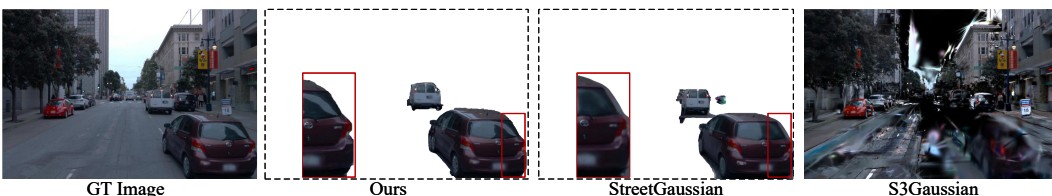

GT Image    Ours    StreetGaussian    S3Gaussian

Figure 4: Visualization of scene decomposition results. Since S$^3$Gaussian is not capable of instance-level decomposition, we present its visualization results on static scene decomposition.

Octree-GS (Ren et al., 2024) are static 3D Gaussian methods. For a fair comparison, we replace the Structure-from-Motion (Schonberger & Frahm, 2016) point clouds in 3DGS with LiDAR point clouds for initialization.

## 4.2 COMPARISONS WITH STATE-OF-THE-ART METHODS

Table 1 presents the comparison results between our method and the baseline approaches. We adopt PSNR, SSIM, and LPIPS as evaluation metrics for rendering quality. To better assess the reconstruction quality of dynamic objects, we following these works (Yan et al., 2023; Huang et al., 2024a; Yang et al., 2023), adopt PSNR* and SSIM*, which are computed by projecting the 3D bounding boxes of dynamic objects onto the 2D image plane and calculate the pixel loss only within the projected boxes. For most metrics, our method achieves the best performance among all the compared approaches. Moreover, due to the multiview consistency between semantic and instance features, our method maintains excellent performance even in static scenes.

Tbable 2 shows the comparison results on the complete Waymo dataset. Compared with baseline methods, our approach does not rely on depth supervision or additional inputs, yet achieves the best performance in PSNR and LPIPS while delivering competitive results in SSIM. In addition, our method supports instance-level control, further enhancing its practical applicability.

Table 3 reports results on the Waymo-Street dataset (Yan et al., 2023). Our method outperforms StreetGaussian without relying on bounding boxes and surpasses S$^3$Gaussian in capturing fine details and rendering quality. For fairness, FPS is measured on a single GPU. StreetGaussian leverages scarce 3D bounding boxes to focus only on parts of the scene, achieving the fastest rendering speed, whereas S$^3$Gaussian interpolates all

Table 3: Quantitative results on Waymo-Street datasets. St-GS denotes StreetGaussian.

| Metrics | 3DGS | MARS | St-GS | S$^3$Gaussian | Ours |
|---|---|---|---|---|---|
| PSNR↑ | 29.64 | 31.37 | 34.96 | 34.61 | **35.09** |
| SSIM↑ | 0.918 | 0.904 | 0.945 | 0.950 | **0.955** |
| LPIPS↓ | 0.117 | 0.246 | 0.068 | 0.050 | **0.047** |
| PSNR*↑ | 16.48 | 23.07 | 25.46 | 25.78 | **26.58** |
| FPS↑ | 227 | 0.68 | **133** | 23 | 37 |

Gaussians on multi-scale Hexplanes with additional MLP, leading to slower rendering.

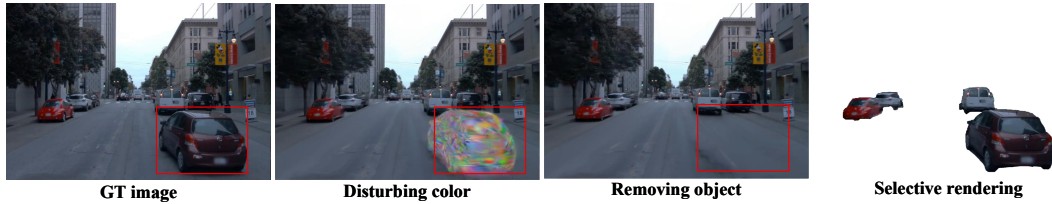

Figure 5: Instance-level visualization results.

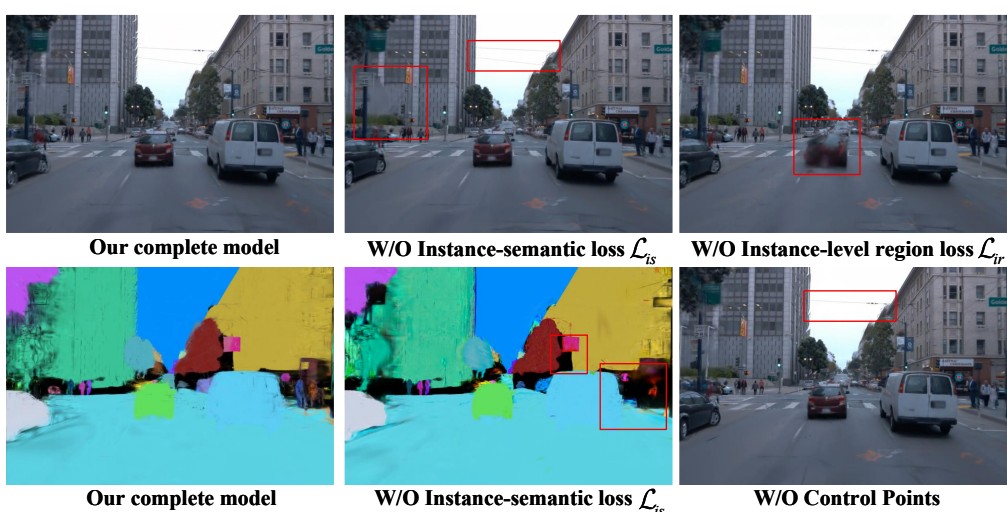

Figure 6: Visual ablation results.

Figure 3 presents qualitative comparisons. S³Gaussian suffers from motion blur when modeling dynamic objects and fails to construct certain static details. StreetGaussian achieves high-fidelity scene reconstruction overall but exhibits noticeable artifacts near object boundaries and omits some static details. 3DGS effectively captures static structures but is unable to handle dynamic elements in the scene. In contrast, our approach produces high-quality renderings with strong multi-view consistency. It significantly reduces artifacts such as ghosting and loss of fine details, demonstrating its effectiveness in reconstructing dynamic scenes.

### 4.3 INSTANCE-LEVEL SCENE VISUALIZATION

Figure 4 presents the scene decomposition results of different methods. The proposed approach achieves the best decomposition without requiring additional 3D annotations. StreetGaussian also demonstrates strong decomposition ability, but its reliance on bounding boxes leads to redundant object boundaries and limits its generalization capability. S³Gaussian distinguishes between static and dynamic regions by modeling the degree of Gaussian attribute deformation; while it can reconstruct dynamic scenes reasonably well, its design is not well suited for precise scene decomposition.

Figure 5 presents instance-level scene editing results that validate the effectiveness of the proposed method. Experimental results show that our method can effectively decompose scenes at the instance level, enabling better reconstruction of dynamic scenes and facilitating downstream applications.

Furthermore, we provide additional experiments on real-world data in the Appendix A.3.

### 4.4 ABLATION AND ANALYSIS

**The influence of Instance-Level Region Loss $\mathcal{L}_{ir}$.** The visualization results reveal that when $\mathcal{L}_{ir} = 0$, the model tends to preserve ghosting artifacts during early training in order to achieve higher rendering quality. This behavior suppresses the output of the deformable network. As a result, the output gradually converges to zero, causing the model to construct a pseudo-dynamic scene.

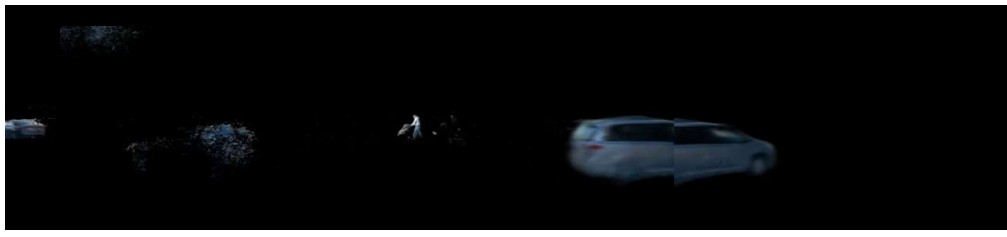

(a) Pseudo labels are perturbed by random modifications and noise.

(b) Pseudo labels are only used for the front camera.

Figure 7: Robustness analysis visualization. We vary the quantity and quality of pseudo-labels to evaluate the robustness. The results demonstrate that the model maintains consistency in unsupervised regions and can achieve coherent outputs across frames even in the presence of label errors.

**The influence of Instance-semantic Loss** $\mathcal{L}_{is}$. Compared to RGB features, semantic features are expected to remain consistent across different viewpoints for the same instance. As shown in Table 4 and Figure 6, both instance and semantic features exhibit strong consistency. This consistency further contributes to an improved spatial distribution of Gaussians, thereby indirectly enhancing rendering quality.

**The influence of control points**. Under this setting, each Gaussian independently computes its offset. In comparison, our method achieves a 2.3× reduction in training time (approximately 3 hours and 40 minutes). Experimental results show that we achieve higher efficiency without compromising rendering quality

Table 4: Ablation studies. "w/o" indicates that the module is removed. Results for "w/o $\mathcal{L}_{ir} = 0$" are omitted since the object does not exhibit actual motion.

| | PSNR↑ | SSIM↑ | LPIPS↓ | PSNR*↑ |
|---|---|---|---|---|
| S³Gaussian | 31.35 | 0.911 | 0.106 | 26.02 |
| w/o $\mathcal{L}_{is}$ | 31.85 | 0.915 | 0.101 | 25.71 |
| w/o $\mathcal{L}_{ir}$ | - | - | - | - |
| w/o control points | 32.68 | 0.922 | 0.098 | 27.03 |
| Complete model | **32.74** | **0.926** | **0.095** | **28.12** |

**Robustness analysis**. We randomly perturb the pseudo-labels, and the results are shown in Figure 7a. The experiments demonstrate that our method produces consistent segmentation results through multi-frame joint optimization. Furthermore, combined with ablation studies indicate that the semantic-instance level loss constrains the spatial distribution of Gaussians belonging to the same instance, thereby generating consistent outputs across multiple frames. We apply pseudo-labels only to the front-facing camera, and the results are shown in Figure 7b. The results demonstrate that although slight positional perturbations occur in some regions, the model can still maintain the spatial positions and motion relationships learned from the front-facing camera across multiple views. Even for distant objects (e.g., vehicles in the left camera and pedestrians in the front camera), the structure and motion remain stable and consistent.

## 5 CONCLUSION

In this paper, we propose Distributed Dynamic Instance Gaussians that decomposes dynamic scene and motion at the instance level. With efficient instance encoding, instance-level control points, and two region-specific losses, we achieve precise and efficient modeling of transient motion. Experimental results show that our method achieves strong performance and generalization, enhances scene reconstruction, and benefits downstream applications.

ETHICS STATEMENT

This work adheres to the ICLR Code of Ethics. Our research focuses on developing fundamental methods for dynamic 3D scene representation and does not involve sensitive personal data or human subjects. We believe our contributions do not raise direct ethical concerns. However, as with all research in 3D perception, potential misuse in surveillance or privacy-infringing applications should be considered.

REPRODUCIBILITY STATEMENT

To ensure reproducibility, we provide detailed experimental settings and training details. All experiments in the main paper are conducted on publicly available datasets, and the SAM pretrained model used is also publicly accessible. The source code and usage documentation will be released upon publication.

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

# A APPENDIX

## A.1 LARGE LANGUAGE MODEL USAGE

This paper only used Large Language Models for language refinement and grammar correction.

## A.2 ADDITIONAL IMPLEMENTATION DETAILS

**Training details**. To initialize instance-level control points and stabilize training, we first reconstruct the static scene over 5,000 iterations. Instance control points are then obtained via farthest point sampling (FPS). If prior instance category information is available, we assign category-specific sampling rates according to Gaussian IDs to better capture motions of different complexities. For simplicity, all experiments in this work employ a uniform sampling rate of 0.01. At this stage, the spatial positions of control points remain suboptimal. To address this, we inherit all attributes from the Gaussian points and optimize the control points for 3,000 iterations without applying any densification. In subsequent stages, only the 3D coordinates of the control points are required, discarding other features. During dynamic scene reconstruction, we feed only the control point coordinates along with the timestamp into MLPs, which simplifies motion modeling. The overall loss is $\mathcal{L} = \mathcal{L}_{color} + \lambda_1 \mathcal{L}_{is} + \lambda_2 \mathcal{L}_{ir}$, where $\lambda_1 = 0.01$ and $\lambda_2 = 0.2$.

**Public Dataset details**. The Waymo-NOTR dataset follows the setup of Yang et al. (2023). We use three frontal cameras, FRONT LEFT, FRONT, and FRONT RIGHT, with images resized to $640 \times 960$ for both training and evaluation. Each sequence contains 100 frames, with every 10th frame reserved for testing and the remaining frames used for training. For the Waymo-Street dataset follows the setup of Yan et al. (2023), we use frontal camera images downscaled to $1066 \times 1600$ to assess monocular reconstruction and novel view synthesis. The length of all sequences is approximately 100 frames long. Every 4th frame is selected for testing, while the rest are used for training.

**Private Dataset details**. The sensor suite we use is shown in Figure 8. It containsa rgb camera (10-60Hz, 720x1280 resolution), a 200Hz Inertial Measurement Unit (IMU), and a triple-antenna Ultra-Wideband (UWB) sensor. Each captured image is processed by the Segment Anything Model to generate pseudo labels. This experiment aims to evaluate the effectiveness of the proposed algorithm in real-world settings, complementing evaluations on public datasets.

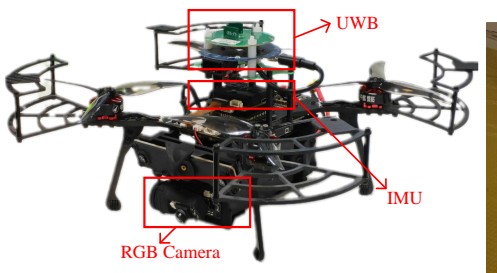 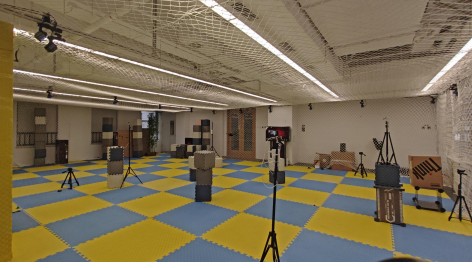

(a) Aerial robot with forward-looking stereo camera,an IMU and a triple-antenna.

(b) Designed experiment scene.

Figure 8: Overview of the experiment setting.

**Distributed System Implementation Details**. We design a distributed system tailored for canonical Gaussians, control points, and the deformation network, building upon the Grendel framework proposed in Zhao et al. (2025). Training Gaussians entails dynamic and non-uniform computation patterns across iterations, which poses significant challenges for efficiency and scalability. To address this, we adopt a hybrid parallelism strategy operating at both the Gaussian and pixel levels to achieve effective load balancing.

Specifically, after initializing Gaussians, we evenly distribute them across GPUs. As reconstruction progresses, regions with varying complexity exhibit different Gaussian growth rates. To mitigate imbalance, we periodically redistribute Gaussians across GPUs following several densification steps. Since the computational cost per pixel varies in rendering, loss calculation, and gradient backpropagation, we measure the runtime per pixel and then reassign image blocks across GPUs such that

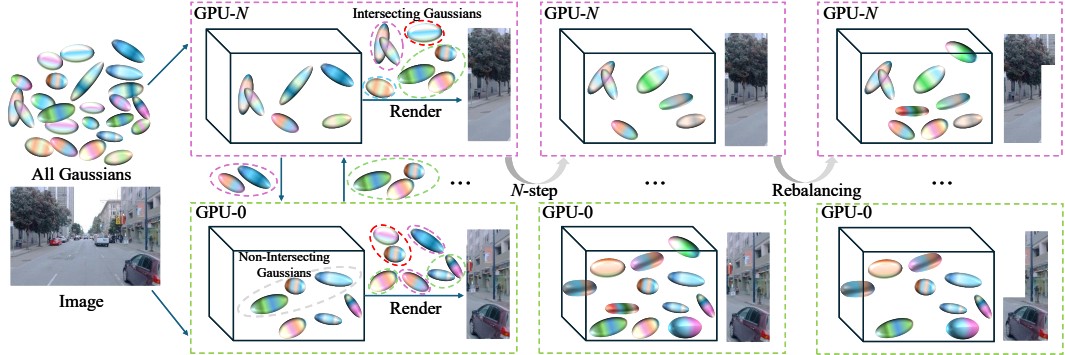

Figure 9: Pipeline of the Distributed System.

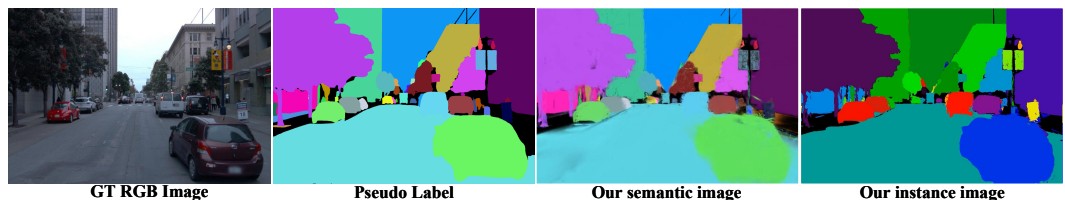

Figure 10: Rendered Results Visualization. The proposed method effectively achieves fine-grained scene understanding. Note that the semantic map refers to the RGB output from SAM, while the instance map represents the RGB visualization of instance ID rendering results.

Table 5: Results of Instance Distinction Method Comparison.

| Methods | time(s)↓ | Memory(Mb)↓ | mIoU ↑ |
|---|---|---|---|
| Gaussian Grouping | 0.2135 | 290.31 | 74.2 |
| Ours | 0.0051 | 259.79 | 71.7 |
| Δ | -97.61% | -10.51% | -3.36% |

all GPUs have equal pixel-level workloads. During each rendering pass, each GPU retrieves the set of intersecting Gaussians within its assigned image region, and then exchanges the relevant Gaussians via sparse all-to-all communication. The distributed framework is illustrated in Figure 9. For sparse control points, which are not involved in the densification process, we distribute them evenly across GPUs at initialization and keep them fixed thereafter. For the deformation network, we adopt PyTorch's data parallelism, where each GPU processes different data batches and the gradients are aggregated for parameter updates. During motion field construction, each GPU first obtains the global control points through all-to-all communication. Then, each Gaussian computes its attribute offsets based on instance encoding and spatial relationships. Finally, control points and Gaussians from all GPUs are gathered to form the complete scene representation.

## A.3 ADDITIONAL EXPERIMENTS

Figure 10 shows the visualized instance-level and semantic results. Notably, the semantic map refers to the RGB result learned by the Gaussians from SAM (which differs from the semantic map typically used in computer vision). The instance map is obtained by rendering the Gaussian instance IDs and mapping the results accordingly. The experimental results demonstrate that the proposed method not only possesses environmental perception capabilities but also effectively distinguishes between different Gaussians. Moreover, the proposed method achieves the panoptic quality (PQ) Kirillov et al. (2019), average precision (AP), and mean IoU (mIoU) of 62.1, 28.6, and 71.7, respectively. Combined with experiments on scene reconstruction and instance-level decomposition, these results demonstrate that our approach can perform instance-level Gaussian distinction, effectively meeting the requirements of dynamic street scenes.

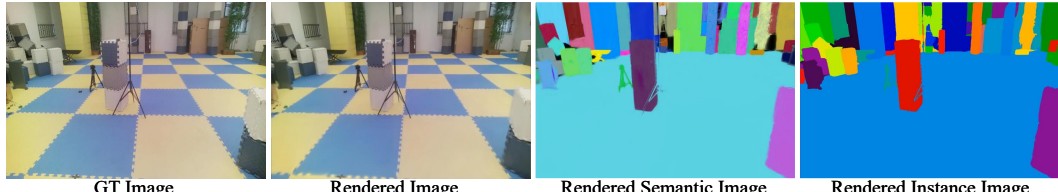

| GT Image | Rendered Image | Rendered Semantic Image | Rendered Instance Image |

Figure 11: Rendering visualization results on real-captured data.

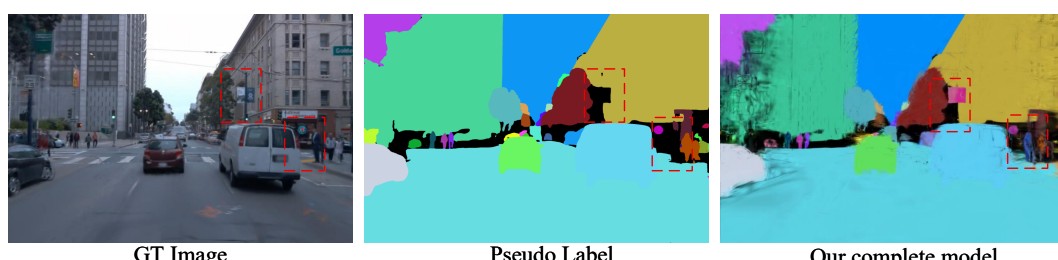

| GT Image | Pseudo Label | Our complete model |

Figure 12: Rendering visualization results on real-captured data.

Table 6: SOTA comparison on the nuPlan (Caesar et al., 2021) dataset.

| Methods | PSNR | SSIM | LPIPS |
|---|---|---|---|
| Deform-Gs (Yang et al., 2024) | 28.10 | 0.880 | 0.143 |
| HUGS (Zhou et al., 2024) | 24.16 | 0.768 | 0.274 |
| Street Gaussians (Yan et al., 2023) | 26.05 | 0.813 | 0.180 |
| OmniRe (Chen et al., 2024) | 26.72 | 0.845 | **0.165** |
| Ours | **29.23** | **0.872** | 0.167 |

Figure 11 shows the reconstruction results on real-captured data. Experimental results demonstrate that our method can effectively reconstruct and understand the scene under various experimental settings (e.g., different FPS and motion speeds, without ground truth poses).

Table 5 shows the quantitative results of Gaussian instance differentiation for different methods. To ensure a fair comparison in complex street scenes, both methods use one million Gaussians for memory computation and instance differentiation. Our method uses 8-dimensional instance features while Gaussian Grouping Ye et al. (2024) uses 16-dimensional features. For Gaussian Grouping, instance classification is implemented with a single-layer MLP producing 256 output classes. Experimental results show that our method achieves comparable segmentation quality while significantly reducing memory usage and runtime. In real-time dynamic scene construction, Gaussian Grouping's per-frame Gaussian selection speed is insufficient to meet practical requirements

We evaluate the consistency of the model's features, and the results are shown in Figure 12. The experiments indicate that although the pseudo labels contain certain inaccuracies, our model successfully propagates correct information from other frames to the current view, enabling globally consistent reconstruction. The perceptual discrepancies arise because the pseudo labels represent discrete instance ID mappings, whereas we visualize the results as continuous RGB images.

To verify the generalizability of the proposed method, we performed experiments on the NuPlan dataset, as illustrated in Table 6. The results show that our method achieves comparable or even superior performance in all metrics, further validating the effectiveness and generalizability of the proposed approach.

We further evaluate the performance gains obtained by scaling the number of GPUs. Increasing the number of GPUs allows the system to employ more Gaussians to represent the scene. Experimental results show that the current setting has not yet reached the quality saturation point of 3DGS. Specifically, when using 3 and 4 GPUs, the PSNR improves by 0.17 and 0.22, respectively. However, as the number of Gaussians increases, the cost of KNN computation grows, and the lack of NVLink further

amplifies GPU communication overhead, leading to a significant slowdown in training. Therefore, we adopt 2 GPUs as the final configuration to balance reconstruction quality and training efficiency.

## A.4 GRADIENT DERIVATION

In this section, we briefly introduce the gradient derivation of the opacity rendering pipeline. Recall that the final pixel opacity is computed as follows:

$$O = \sum_{i \in \mathcal{N}} \alpha \alpha_i' T_i, \text{ where } T_i = \prod_{j=1}^{i-1}(1 - \alpha_j'), \tag{6}$$

where the final opacity $\alpha_i'$ is the multiplication result of the learned opacity $\alpha$ and the Gaussian:

$$\alpha_i' = \alpha_i \exp(\frac{1}{2}(x' - \mu_i')^T \Sigma_i^{2D-1}(x' - \mu_i')), \tag{7}$$

where , $x'$ and $\mu_i'$ are coordinates in the projected space, and $\alpha_i^{2D}$ is the 2D covariance. Note the coupling between $\alpha_i$ and $\alpha_i'$. Equation 6 can be reformulated as:

$$O = \sum_{i=1}^{k} \alpha_i \alpha_i' \prod_{j=1}^{i-1}(1 - \alpha_j') + \sum_{i=k+1}^{N} \alpha_i \alpha_i' \prod_{j=1}^{i-1}(1 - \alpha_j'). \tag{8}$$

As a result, we find the gradients for the variable a as follows:

$$\frac{dO}{d\alpha_i'} = 2\alpha_i T_i - \frac{1}{(1-\alpha_i')} \sum_{j=i+1}^{N} \alpha_j \alpha_j' T_j$$

$$= T_i(2\alpha_i - \sum_{j=i+1}^{N} \alpha_j \alpha_j' \prod_{l=i+1}^{j-1}(1 - \alpha_l')) \tag{9}$$

where the coefficient 2 in the term $2\alpha_i T_i$ arises from the coupling between $\alpha_i$ and $\alpha_i'$. Leveraging the chain rule, the final opacity gradient is computed in coordination with other the rendering pipeline. Therefore, we only derive the expression up to $dO/d\alpha_i'$ here.

## A.5 DISCUSSION

The core of our approach lies in jointly optimizing instance attributes using 2D instance ID masks and instance-segmented RGB images. This allows fast differentiation and control of Gaussian instances without MLPs. Meanwhile, we disentangle scene and motion by enforcing instance motion consistency, spatial continuity, and local rigidity, and employ two instance-level losses to optimize both the spatial distribution of Gaussians and the feature consistency. Compared to standard rasterization, we additionally incorporate both an instance rendering pipeline and an opacity rendering pipeline, and reformulate the gradient computation for opacity.

**Difference from Bounding-Box–Based Methods**. First, in terms of data cost, our method relies solely on model-generated pseudo-labels to achieve high-quality instance-level disentanglement of scene and motion, whereas bounding-box–based methods require costly human-annotated 3D information. Second, regarding motion complexity, bounding-box–based methods typically assume rigid object motion. For instance, StreetGaussian explicitly notes that it cannot handle non-rigid dynamic objects such as walking pedestrians. While this assumption captures local motion patterns, real-world motion is far more complex than a single rigid pattern. Our method extends the idea of local rigidity by introducing multiple control points to jointly drive the motion of each entity, thereby supporting a wider range of motion patterns. Moreover, if additional semantic information is available, the sampling rate of control points can be adjusted to model motions of varying complexity. Third, in terms of category flexibility, our method can explicitly model any object detected by the Segment Anything Model, whereas bounding-box–based approaches are limited to fixed categories with reliable 3D tracking. Additionally, our instance-level control operates at the Gaussian level, while bounding-box–based approaches operate at the box level, providing more flexible instance

boundaries. Finally, the multi-view consistency of instance features inherent to our approach not only facilitates downstream applications but also benefits the optimization of the Gaussian representations themselves.

**Limitations**. While enabling instance-level motion decoupling in dynamic scenes, DDI-GS still has certain limitations. First, we do not model lighting, which may lead to visual inconsistencies. Second, like many non-feed-forward Gaussian methods, the current approach relies on per-scene optimization. Addressing these non-trivial challenge requires dedicated efforts beyond the scope of our current work. In future work, we will investigate lighting modeling and feed-forward architectures, explore the integration of our approach with part-level decomposition, and conduct more extensive experiments on datasets such as Waymo and NuPlan, comparing with methods like OmniRe Chen et al. (2024), MTGS (Li et al., 2025a), and Desire-GS (Peng et al., 2025). In addition, we plan to integrate Gsplat into the distributed implementation to achieve higher efficiency. In addition, we plan to integrate Gsplat (Ye et al., 2025) into the distributed implementation to achieve higher efficiency.

