# OpenReview forum: "DDI-Gaussian: Distributed Dynamic Instance Gaussian for Autonomous Driving"
_ICLR.cc/2026/Conference — ICLR 2026 Conference Withdrawn Submission_

### Official Review · Reviewer_KAej · 2025-10-26

**Soundness:** 2
**Presentation:** 2
**Contribution:** 2
**Rating:** 4
**Confidence:** 4

**Summary:**

This paper presents DDI-Gaussian, a new framework for dynamic 3D Gaussian scene reconstruction, especially for street scene reconstruction. Main contribution lies in the instance control compared to previous methods. In addition, the method is implemented in a distributed multi-GPU system and leverages 2D pseudo-labels from SAM to avoid costly 3D annotations. Experiments on Waymo-NOTR and Waymo-Street datasets demonstrate improved scene reconstruction and rendering quality.

**Strengths:**

1. Instance-level control is essential for the self-supervised method like S3Gaussian.
2. Distributed training implementation is also practical for street scene reconstruction.

**Weaknesses:**

1. Limited novelty: the core idea seems a mixed of existing semantic 3DGS works. The instance control can be implemented by panoptic supervision and the vanilla rasterization can be changed into gsplat for better distributed training pratice.
2. Lack of baselines: OmniRe (ICLR2025), SplatAD(CVPR2025), DeSiRe-GS (CVPR2025), EMD (ICCV2025).

**Questions:**

1. Does the method scale efficiently beyond two GPUs?
2. Could instance-level encoding be extended to support part-level or semantic decomposition?

---

> ### Author Response · Authors · 2025-11-16
>
> We would like to thank you for your thorough review and the valuable time you have devoted. In the following, we address your comments and concerns point by point. All corresponding revisions have been incorporated into the manuscript and are highlighted in red for your convenience.\
> \
> 1.**We have added comparisons with the latest methods in Table 1 (corresponding to Table 2 in the paper)**. The results show that our method achieves strong performance without requiring extra inputs or depth supervision, while enabling finer instance-level control. To better clarify our contributions, we refined the descriptions in the abstract and introduction. We added **comparisons with various SOTA methods** in Table 2, incorporated **robustness analysis** in Figure 7, and included **segmentation performance analysis**, **feature consistency analysis** and **a comparison of the Gaussian identification cost** in rebuttal.\
> **Table 1: Quantitative comparison of scene reconstruction results on the Waymo dataset.  D$_{\text{opt}}$ denotes depth optimization. “Box” indicates methods that utilize bounding boxes for dynamic modeling. SR and NVS indicate Scene Reconstruction and Novel View Synthesis, respectively**
>
> | Methods   | Venue     | Box  | D$_{\text{opt}}$ | PSNR↑ (SR) | SSIM↑ (SR) | LPIPS↓ (SR) | PSNR↑ (NVS) | SSIM↑ (NVS) | LPIPS↓ (NVS) |
> | --------- | --------- | ---- | ---------------- | ------------- | ------------- | -------------- | ----------- | ----------- | ------------ |
> | 4D-GS     | CVPR2024  | ✗    | ✗                | 31.01         | 0.890         | 0.205          | 28.16       | 0.870       | 0.219        |
> | Deform-GS | CVPR2024  | ✗    | ✗                | 32.66         | 0.912         | 0.158          | 30.19       | 0.894       | 0.167        |
> | Octree-GS | TPAMI2025 | ✗    | ✗                | 33.63         | 0.936         | 0.103          | 32.13       | **0.927**   | 0.107        |
> | Desire-GS | CVPR2025  | ✓    | ✓                | 34.55         | 0.926         | 0.108          | 32.56       | 0.919       | 0.119        |
> | St-GS     | ECCV2024  | ✓    | ✗                | 35.66         | 0.936         | 0.109          | 30.87       | 0.882       | 0.139        |
> | S3-GS     | ICCV2025  | ✗    | ✓                | 33.44         | 0.927         | 0.118          | 31.62       | 0.917       | 0.124        |
> | OmniRe    | ICLR2025  | ✓    | ✓                | 34.95         | **0.943**     | 0.114          | 31.79       | 0.905       | 0.125        |
> | **Ours**  | N/A       | ✗    | ✗                | **36.12**     | 0.942         | **0.075**      | **34.49**   | **0.927**   | **0.082**    |
>
> \
> 2. **Novelty**. The core of our method does not lie in distributed training, so it is not listed separately as a contribution. Distributed training is included only to allow more researchers to perform dynamic scene reconstruction when single-GPU memory is limited. Unlike existing semantic methods, **the core of our approach** lies in jointly optimizing instance attributes using 2D instance ID masks and instance-segmented RGB images. This allows fast differentiation and control of Gaussian instances **without MLPs**. Meanwhile, we **disentangle scene and motion by enforcing instance motion consistency, spatial continuity, and local rigidity**, and employ **two instance-level losses** to optimize both the spatial distribution of Gaussians and the feature consistency. Compared to standard rasterization, we additionally incorporate both an instance rendering pipeline and an opacity rendering pipeline, and **reformulate the gradient computation for opacity**.\
> \
> 3.**Our method can run on any number of GPUs**, and we evaluate the performance gains obtained by scaling the number of GPUs in Appendix A.3. Specifically, when using 3 and 4 GPUs, the PSNR improves by 0.17 and 0.22, respectively. However, as the number of Gaussians increases, the cost of KNN computation grows, and the lack of NVLink further amplifies GPU communication overhead, leading to a significant slowdown in training. Therefore, we adopt 2 GPUs as the final configuration to balance reconstruction quality and training efficiency. The main purpose of the distributed implementation is to allow researchers to reproduce our work even when single-GPU memory is limited.\
> \
> 4.**Part-Level  Decomposition**. The instance-level encoding in our method provides a new approach for encoding and optimizing Gaussian spatial positions, which **can be naturally extended to part-level representations**—either further using part-level labels or clustering based on spatial positions. **Our method can retain the original instance features without retraining**. For example, we can design the feature as (N+M)-dimensional, with the first N dimensions representing the instance and the remaining M dimensions refined into different parts according to labels or spatial positions. We greatly appreciate this excellent idea, which will inspire much of our future work.

---

> > ### Comment · Reviewer_KAej · 2025-11-17
> > **Reply to the rebuttal**
> >
> > Thanks for your rebuttal!
> >
> > The rebuttal mostly addresses my concerns. But I still have two questions:
> > 1. It seems that your code base is self-supervised based street Gaussian reconstruction. But in practice, box supervised based   methods allow a better instance-level control and scene edits for data augmentation. So why not directly enhence box supervised based methods such as OmniRe or SplatAD with instance grouping strategy?
> > 2. And why did you implement the distributed training in the vanilla 3DGS codebase but not begin with Gsplat? It may allow a faster optimization speed.

---

> > > ### Author Response · Authors · 2025-11-17
> > >
> > > We sincerely thank you again for taking the time to provide your professional feedback. We will address your questions point by point.\
> > > \
> > > **1.Difference from bounding-box–based methods**. Our implementation is built upon Grendel-Gs[1], as further clarified in the next response. Our method differs from bounding-box–based approaches such as OmniRe and StreetGaussian in four aspects. **a) Data cost**. Our method only relies on model-generated pseudo-labels to achieve high-quality instance-level disentanglement of scene and motion. In contrast, Box methods require more expensive human-annotated 3D information.  **b) Motion Complexity.** Bounding-box–based methods typically assume that objects undergo rigid motion. For example, StreetGaussian explicitly states that it cannot handle non-rigid dynamic objects such as walking pedestrians. While this assumption captures local motion patterns, real-world motion is far more complex than a single rigid pattern. Our method inherits the idea of local rigidity and extends it by using multiple control points to jointly drive the motion of each entity, enabling a wider range of motion patterns. If additional semantic information is available, the sampling rate of control points can be adjusted to model motions with different levels of complexity. **c) Unknown categories**. Our method can explicitly model any object detected by the Segment Anything Model (SAM) [2] in the scene, whereas box-based approaches can only control fixed categories that have reliable 3D tracking. Moreover, our instance-level control operates at the Gaussian level, while box-based approaches operate at the box level, giving us much more flexible instance boundaries. **d) The inherent multi-view consistency of instance features** not only facilitates downstream applications but also benefits the optimization of Gaussians themselves. **We greatly appreciate your valuable feedback, which is very insightful. Combining this with part-level decomposition would be a very interesting idea.**
> > > \
> > > \
> > > 2. **Code implementation and distribution**. Our implementation is built upon Grendel-Gs [1], which is specifically designed for distributed Gaussians but only supports static scenes. To adapt it to dynamic street scenes and instance-level control, we extend it to a distributed framework incorporating Gaussians, sparse control points, and deformation networks, and adding instance and opacity map rendering pipelines. Distributed training is not the core focus of this work and is intended to allow more researchers to reproduce our results even with limited single-GPU memory. Therefore, we did not highlight it as a contribution in the introduction. **We thank you for this insightful comment. In future work, we will study how to better integrate the Grendel-GS framework with Gsplat to achieve faster optimization.**\
> > > \
> > > [1] Hexu Zhao, Haoyang Weng, Daohan Lu, Ang Li, Jinyang Li, Aurojit Panda, and Saining Xie. On scaling up 3d gaussian splatting training. In European Conference on Computer Vision, pp. 14–36. Springer, 2025
> > >
> > > [2] Alexander Kirillov, Eric Mintun, Nikhila Ravi, Hanzi Mao, Chloe Rolland, Laura Gustafson, Tete Xiao, Spencer Whitehead, Alexander C Berg, Wan-Yen Lo, et al. Segment anything. In Proceedings of the IEEE/CVF international conference on computer vision, pp. 4015–4026, 2023.

---

> ### Comment · Reviewer_KAej · 2025-11-17
> **Reply to the comment**
>
> Thanks for your timely reply!
>
> 1. Actually, the motivation "expensive human-annotated 3D information" appears in every self-supervised based method. In a practical pipeline, simply integrating the open-vocabulary detector can address this limitation and achieve promising results (maybe with noise). But I acknowledge that this paper investigates the effect of introducing semantic information. I recommend that authors carefully refine the initial version of the paper to avoid confusion.
>
> 2. I don't think Grendel-GS is a good choice for project startup, which is dedicated to large-scale urban scene reconstruction. Thus, I also strongly recommend that authors integrate the current pipeline into Gsplat,  which may reduce extra code design.
>
> 3. Additionally, I recommend that authors implement experiments on NuPlan with multi-traversal sensor data to test the performance on lane change settings (such as MTGS: https://github.com/OpenDriveLab/MTGS). It is more crucial for street scene reconstruction rather than novel view synthesis on the same lane.
>
> Finally, I tend to raise my score for the valuable attempt in this paper. But I also believe this paper needs to be refined to meet the acceptance standards of ICLR.

---

> > ### Author Response · Authors · 2025-11-17
> >
> > **We sincerely thank you for taking the time to help us further improve our paper.** We will address your final concerns point by point.\
> > \
> > 1.**Paper description**. We have further strengthened the presentation in the revised version to help readers better understand the advantages and significance of our approach. **During the rebuttal process,  we will continue refining the  to meet the acceptance standards of ICLR.**\
> > \
> > 2.**Integration with Gsplat**. Your suggestion is very constructive, and we will include it in our TODO list, which will also facilitate future extensions of our work.\
> > \
> > 3.**NuPlan dataset**. NuPlan is a recently released, high-quality dataset specifically designed for complex urban street scenarios, making it highly suitable for dynamic Gaussian scene modeling in autonomous driving research. This dataset aligns very well with our current and future work. This is an excellent work and closely related to our research direction. **We will cite this paper in the discussion section of the appendix and begin experiments on this dataset**.\
> > \
> > **We again sincerely thank you for taking the time to provide valuable suggestions for our work.**

---

> ### Author Response · Authors · 2025-11-19
>
> **We sincerely thank the reviewer for the valuable suggestion**, which further improved the completeness of our experiments. Following the comment, we conducted additional evaluations on the NuPlan dataset. **The results show that** our method achieves comparable or even superior performance across all metrics, further validating the effectiveness and generalization capability of the proposed approach.\
> \
> **Table 1: Quantitative comparison of scene reconstruction results on the NuPlan dataset.**
>
> | Methods          | PSNR↑      | SSIM↑      | LPIPS ↓    |
> | ---------------- | --------- | --------- | --------- |
> | Deform-GS [1]        | 28.10     | 0.880     | 0.143     |
> | HUGS [2]             | 24.16     | 0.768     | 0.274     |
> | Street Gaussians [3] | 26.05     | 0.813     | 0.180     |
> | OmniRe [4]           | 26.72     | 0.845     | **0.165** |
> | Ours             | **29.23** | **0.872** | 0.167     |
>
> \
> [1] Ziyi Yang, Xinyu Gao, Wen Zhou, Shaohui Jiao, Yuqing Zhang, and Xiaogang Jin. Deformable 3d gaussians for highfidelity monocular dynamic scene reconstruction. In CVPR, 2024. 6, 7, 8, 12\
> [2] Hongyu Zhou, Jiahao Shao, Lu Xu, Dongfeng Bai, Weichao Qiu, Bingbing Liu, Yue Wang, Andreas Geiger, and Yiyi
> Liao. Hugs: Holistic urban 3d scene understanding via gaussian splatting. In CVPR, 2024. 2, 6, 7, 12\
> [3] Yunzhi Yan, Haotong Lin, Chenxu Zhou, Weijie Wang, Haiyang Sun, Kun Zhan, Xianpeng Lang, Xiaowei Zhou,
> and Sida Peng. Street gaussians: Modeling dynamic urban scenes with gaussian splatting. In ECCV, 2024. 2, 4, 5, 6, 7, 8, 12, 13\
> [4] Ziyu Chen, Jiawei Yang, Jiahui Huang, Riccardo de Lutio, Janick Martinez Esturo, Boris Ivanovic, Or Litany, Zan Gojcic, Sanja Fidler, Marco Pavone, et al. Omnire: Omni urban scene reconstruction. arXiv preprint arXiv:2408.16760, 2024

---

### Official Review · Reviewer_k2AD · 2025-10-29

**Soundness:** 2
**Presentation:** 2
**Contribution:** 2
**Rating:** 4
**Confidence:** 4

**Summary:**

DDIG assigns a compact multi-hot instance feature to each Gaussian, enabling self-supervised modeling of different instances within a scene without requiring 3D instance annotations. Additionally, distributed training is employed to accelerate the overall training process.

**Strengths:**

- The method does not rely on 3D instance annotations, which reduces annotation costs for large-scale applications.

 - Distributed training significantly accelerates training speed.

**Weaknesses:**

- The paper lacks comparisons with state-of-the-art methods (e.g., OmniRe, BezierGS).

 - The approach does not clearly demonstrate how deformable traffic participants are modeled.

 - For modern end-to-end autonomous driving systems, the explicit semantic modeling in DDIG may be less critical. Moreover, noticeable floaters appear around object boundaries in the semantic renderings.

**Questions:**

- Although the method is annotation-free, comparisons with state-of-the-art approaches that use 3D annotations (e.g., OmniRe, BezierGS) are still necessary, since 3D annotations are not prohibitively expensive relative to the potential reconstruction quality loss.

 - How does the proposed method handle deformable traffic participants?

 - The advantages of DDIG over existing methods should be presented more clearly (e.g., through RGB renderings or instance-level rendering videos). Currently, the improvement over StreetGaussian is not very evident—for instance, in Figure 3, the ground surface appears blurrier than in StreetGaussian.

---

> ### Author Response · Authors · 2025-11-16
>
> **We would like to thank you for your thorough review and the valuable time you have devoted**. In the following, we address your comments and concerns point by point. All corresponding revisions have been incorporated into the manuscript and are highlighted in red for your convenience.\
> \
> 1.**We have added comparisons with the latest methods in Table 1 (corresponding to Table 2 in the paper) and Table 2 (corresponding to Table 6 in the paper)**. The results in Table 1 show that our method achieves strong performance without requiring extra inputs or depth supervision, while providing finer instance-level control. The results in Table 2 further validate the effectiveness and generalizability of the proposed approach. To better clarify our contributions, we refined the descriptions in the abstract and introduction. Additionally, for the revised paper, we incorporated **robustness analysis** in Figure 7, and included **segmentation performance analysis**, **feature consistency analysis** and **a comparison of the Gaussian identification cost** in Appendix A.3.\
> \
> **Table 1: Quantitative comparison of scene reconstruction results on the Waymo dataset.  D$_{\text{opt}}$ denotes depth optimization. “Box” indicates methods that utilize bounding boxes for dynamic modeling. SR and NVS indicate Scene Reconstruction and Novel View Synthesis, respectively**
> | Methods   | Venue     | Box  | D$_{\text{opt}}$ | PSNR↑ (SR) | SSIM↑ (SR) | LPIPS↓ (SR) | PSNR↑ (NVS) | SSIM↑ (NVS) | LPIPS↓ (NVS) |
> | --------- | --------- | ---- | ---------------- | ------------- | ------------- | -------------- | ----------- | ----------- | ------------ |
> | 4D-GS[1]     | CVPR2024  | ✗    | ✗                | 31.01         | 0.890         | 0.205          | 28.16       | 0.870       | 0.219        |
> | Deform-GS[2] | CVPR2024  | ✗    | ✗                | 32.66         | 0.912         | 0.158          | 30.19       | 0.894       | 0.167        |
> | Octree-GS[3] | TPAMI2025 | ✗    | ✗                | 33.63         | 0.936         | 0.103          | 32.13       | **0.927**   | 0.107        |
> | Desire-GS[4] | CVPR2025  | ✓    | ✓                | 34.55         | 0.926         | 0.108          | 32.56       | 0.919       | 0.119        |
> | St-GS[5]     | ECCV2024  | ✓    | ✗                | 35.66         | 0.936         | 0.109          | 30.87       | 0.882       | 0.139        |
> | S3-GS[6]     | ICCV2025  | ✗    | ✓                | 33.44         | 0.927         | 0.118          | 31.62       | 0.917       | 0.124        |
> | OmniRe[7]    | ICLR2025  | ✓    | ✓                | 34.95         | 0.943     | 0.114          | 31.79       | 0.905       | 0.125        |
> | BezierGs[12]    | ICCV2025  |✗   | ✓                |35.77       | **0.954**     | 0.076         | 33.74       | **0.935**       | **0.073**        |
> | **Ours**  | N/A       | ✗    | ✗                | **36.12**     | 0.942         | **0.075**      | **34.49**   | 0.927   | 0.082    |
>
> \
> **Table 2: Quantitative comparison of scene reconstruction results on the NuPlan dataset.**
> | Methods    | PSNR↑      | SSIM↑     | LPIPS ↓    |
> | ------------- | --------- | --------- | --------- |
> | Deform-GS[2]       | 28.10     | 0.880     | 0.143     |
> | HUGS[8]            | 24.16     | 0.768     | 0.274     |
> | Street Gaussians[5] | 26.05     | 0.813     | 0.180     |
> | OmniRe[7]           | 26.72     | 0.845     | **0.165** |
> | Ours             | **29.23** | **0.872** | 0.167     |
>
> \
> 2.**Handling Deformable Traffic Participants.** Unlike StreetGaussian, which targets only vehicles, our method fits the rendered Gaussian instance features to 2D instance segmentation pseudo labels, enabling instance-level control over **arbitrary objects** in the scene.  For both humans (e.g., as shown in Fig. 7 of the paper) and vehicles, motion is realized through sparse instance-level control points, using the same implementation.  **Furthermore**, if additional semantic information (e.g., object categories) is available, different control point sampling rates can be applied to better accommodate the varying complexity of object motions.\
> \
> 3.We provide **videos** of instance-level rendering and control in the **Supplementary Material** and in the **anonymous repository mentioned in the paper abstract** (link: https://anonymous.4open.science/r/DDI-GS-750E/), showcasing the performance advantages of our method and its potential for practical applications.\
> \
> 4. **The significance and prospects of our work**.Explicitly distinguishing 3D entities enables accurate perception and tracking of individual dynamic objects, ensures spatio-temporal consistency across frames, and allows local motion to be effectively decomposed and controlled. This instance-level modeling not only provides interpretable and robust representations for dynamic scene reconstruction  but also facilitates downstream tasks such as prediction, planning, and scene reconstruction.

---

> > ### Author Response · Authors · 2025-11-23
> >
> > 5.**Object boundary**. From Figure 4 and the provided rendering video, it can be observed that, compared to existing methods, our approach achieves finer reconstruction along object boundaries without requiring additional input. We employ two complementary strategies: learning instance IDs and instance-segmented RGB images simultaneously. The instance IDs are used for Gaussian selection, while the RGB images of the instance segmentation map, combined with the corresponding loss, are used to optimize the spatial distribution of Gaussians. The blur observed in the semantic maps and reconstructed semantic maps is due to the fact that semantic maps are generated from discrete instance ID mappings, whereas the visualized results show rendered RGB continuous values. This is further discussed in line 896 of the appendix to help readers better understand our work.\
> > \
> > 6.**Difference from bounding-box–based methods**. Our method differs from bounding-box–based approaches such as OmniRe and StreetGaussian in four aspects. **a) Data cost**. Our method only relies on model-generated pseudo-labels to achieve high-quality instance-level disentanglement of scene and motion. In contrast, Box methods require more expensive human-annotated 3D information. **b) Motion Complexity**. Bounding-box–based methods typically assume that objects undergo rigid motion. For example, StreetGaussian explicitly states that it cannot handle non-rigid dynamic objects such as walking pedestrians. While this assumption captures local motion patterns, real-world motion is far more complex than a single rigid pattern. Our method inherits the idea of local rigidity and extends it by using multiple control points to jointly drive the motion of each entity, enabling a wider range of motion patterns. If additional semantic information is available, the sampling rate of control points can be adjusted to model motions with different levels of complexity. **c) Unknown categories**. Our method can explicitly model any object detected by the Segment Anything Model (SAM) [9] in the scene, whereas box-based approaches can only control fixed categories that have reliable 3D tracking. Moreover, our instance-level control operates at the Gaussian level, while box-based approaches operate at the box level, giving us much more flexible instance boundaries. **d) The inherent multi-view consistency of instance features** not only facilitates downstream applications but also benefits the optimization of Gaussians themselves. We greatly appreciate your valuable feedback, which is very insightful. Combining this with part-level decomposition would be a very interesting idea.\
> > \
> > [1] Guanjun Wu, Taoran Yi, Jiemin Fang, Lingxi Xie, Xiaopeng Zhang, Wei Wei, Wenyu Liu, Qi Tian, and Xinggang Wang. 4d gaussian splatting for real-time dynamic scene rendering. In Proceedings of the IEEE/CVF conference on computer vision and pattern recognition, pp. 20310–20320, 2024.\
> > [2] Ziyi Yang, Xinyu Gao, Wen Zhou, Shaohui Jiao, Yuqing Zhang, and Xiaogang Jin. Deformable 3d gaussians for high-fidelity monocular dynamic scene reconstruction. In Proceedings of the IEEE/CVF conference on computer vision and pattern recognition, pp. 20331–20341, 2024.\
> > [3] Kerui Ren, Lihan Jiang, Tao Lu, Mulin Yu, Linning Xu, Zhangkai Ni, and Bo Dai. Octree- gs: Towards consistent real-time rendering with lod-structured 3d gaussians. arXiv preprint arXiv:2403.17898, 2024.\
> > [4] Chensheng Peng, Chengwei Zhang, Yixiao Wang, Chenfeng Xu, Yichen Xie, Wenzhao Zheng, Kurt Keutzer, Masayoshi Tomizuka, and Wei Zhan. Desire-gs: 4d street gaussians for static-dynamic decomposition and surface reconstruction for urban driving scenes. In Proceedings of the Computer Vision and Pattern Recognition Conference, pp. 6782–6791, 202\
> > [5] Yunzhi Yan, Haotong Lin, Chenxu Zhou, Weijie Wang, Haiyang Sun, Kun Zhan, Xianpeng Lang, Xiaowei Zhou, and Sida Peng. Street gaussians: Modeling dynamic urban scenes with gaussian splatting. In ECCV, 2024. 2, 4, 5, 6, 7, 8, 12, 13\
> > [6] Nan Huang, Xiaobao Wei, Wenzhao Zheng, Pengju An, Ming Lu, Wei Zhan, Masayoshi Tomizuka, Kurt Keutzer, and Shanghang Zhang. Sgaussian: Self-supervised street gaussians for autonomous driving. CoRR, 2024a.\
> > [7] Ziyu Chen, Jiawei Yang, Jiahui Huang, Riccardo de Lutio, Janick Martinez Esturo, Boris Ivanovic, Or Litany, Zan Gojcic, Sanja Fidler, Marco Pavone, et al. Omnire: Omni urban scene reconstruction. arXiv preprint arXiv:2408.16760, 2024\
> > [8] Hongyu Zhou, Jiahao Shao, Lu Xu, Dongfeng Bai, Weichao Qiu, Bingbing Liu, Yue Wang, Andreas Geiger, and Yiyi Liao. Hugs: Holistic urban 3d scene understanding via gaussian splatting. In CVPR, 2024. 2, 6, 7, 12\
> > [9]  Alexander Kirillov, Eric Mintun, Nikhila Ravi, Hanzi Mao, Chloe Rolland, Laura Gustafson, Tete Xiao, Spencer Whitehead, Alexander C Berg, Wan-Yen Lo, et al. Segment anything. In Proceedings of the IEEE/CVF international conference on computer vision, pp. 4015–4026, 2023.

---

> > > ### Author Response · Authors · 2025-11-26
> > >
> > > Beyond the comparison with bounding-box–based methods discussed in Response (6),  we further **compare with instance-based Gaussian approaches** to highlight the advantages of our method.
> > >
> > > \
> > > (7) **Differences from instance-based methods**. We compare our approach with **the latest hard-encoded method OpenGaussian [11]**  and the **soft-encoded method Gaussian-Grouping [10]** . Instance-based approaches such as Gaussian-Grouping assign a high-dimensional vector to each Gaussian and classify it with an MLP. **In contrast**, our method substantially reduces both memory usage and runtime while achieving comparable segmentation quality, **as shown in Table 3**. Methods such as OpenGaussian map high-dimensional instance hard encodings and spatial features to a finite set of prototype vectors. **In contrast**, our approach supports **plug-and-play** category expansion simply by increasing the feature dimensionality, without any retraining. Furthermore, feature-space consistency is achieved solely through the combination of instance RGB maps and an instance semantic loss, eliminating the need for a complex two-level codebook design.
> > >
> > > \
> > > **Table3: Results of Instance Distinction Method Comparison**.
> > > | Methods               | time(s) ↓   | Memory(Mb) ↓ | mIoU ↑     |
> > > | --------------------- | ----------- | ------------ | ---------- |
> > > | Gaussian Grouping [7] | 0.2135      | 290.31       | 74.2       |
> > > | Ours                  | 0.0051      | 259.79       | 71.7       |
> > > | Δ                     | **-97.61%** | **-10.51%**  | **-3.36%** |
> > >
> > > \
> > > **We sincerely appreciate your valuable feedback and would be glad to further discuss any concerns. We hope that our clarifications adequately address your questions**.
> > >
> > > \
> > > [10] Mingqiao Ye, Martin Danelljan, Fisher Yu, and Lei Ke. Gaussian grouping: Segment and edit anything in 3d scenes. In European Conference on Computer Vision, pp. 162–179. Springer, 2024.\
> > > [11] Yanmin Wu, Jiarui Meng, Haijie Li, Chenming Wu, Yahao Shi, Xinhua Cheng, Chen Zhao, Haocheng Feng, Errui Ding, Jingdong Wang, and Jian Zhang. Opengaussian: Towards point-level 3d gaussian-based open vocabulary understanding. In Proceedings of the Advances in Neural Information Processing Systems (NeurIPS), pp. 19114–19138, 2024b.

---

> ### Author Response · Authors · 2025-11-28
>
> BezierGs [12] is an excellent work. We have included it as a baseline in Table 1 of Response (1) and **will cite and discuss it in the revised manuscript**. Below, we clarify the differences between our proposed method and BezierGs:
>
> \
> **(1) Modeling Non-Rigid Objects.** BezierGs adopts a bounding-box–like strategy by using optimizable Bézier curves to represent point offsets, which can only partially address non-rigid dynamics.  In contrast, our method leverages instance features to distinguish arbitrary objects in the scene and samples a sparse set of instance control points. The overall motion of each instance is then decomposed into a joint representation of multiple local control points. This design allows us to effectively handle highly non-rigid objects such as pedestrians, as demonstrated in Figure 7 of the revised manuscript.
>
> \
> **(2) Handling Artifacts.** BezierGs mitigates artifacts by enforcing inter-curve consistency, encouraging the magnitude of offsets and centers to remain coherent. In contrast,our method renders Gaussians belonging to the same instance independently and imposes an **instance-level regional loss** to regulate the rendering scope, thereby constraining their spatial positions. Furthermore, we leverage the inherent multi-view consistency and spatial proximity of instance features to further optimize the spatial distribution of the Gaussians.
>
> \
> **(3) Fine-grained Scene Understanding.** While both methods utilize segmentation models, our approach not only fulfills the requirements of downstream applications for scene understanding but also exploits this fine-grained understanding to enhance scene reconstruction and novel view synthesis. This provides a unified and more expressive scene representation.
>
> \
> **We sincerely thank you again for the time and effort. We would be very glad to further discuss any remaining concerns during the rebuttal period. We hope that our responses have addressed your questions.**
>
> \
> [12] Ma Z, Jiang J, Chen Y, et al. BézierGS: Dynamic Urban Scene Reconstruction with Bézier Curve Gaussian Splatting[J]. arXiv e-prints, 2025: arXiv: 2506.22099.

---

### Official Review · Reviewer_qcpz · 2025-10-31

**Soundness:** 3
**Presentation:** 3
**Contribution:** 2
**Rating:** 4
**Confidence:** 5

**Summary:**

1. This paper studies dynamic 3D scene reconstruction for autonomous driving. It focuses on the problem that current 3D Gaussian Splatting (3DGS) methods lack instance-level understanding in large-scale street scenes.
2.  The authors propose Distributed Dynamic Instance Gaussian (DDI-Gaussian), introducing (1) Multi-hot instance encoding, where each Gaussian is assigned a learnable multi-hot vector to directly distinguish different instances, supervised by masks generated from the Segment Anything Model. (2) Sparse Instance Control Points, where each instance selects a small set of control points using FPS. These control points model sparse motion fields through a lightweight MLP, and dense motion fields are obtained via KNN and linear blend skinning interpolation.
3.  Experiments are conducted on Waymo-NOTR and Waymo-Street. The proposed approach outperforms prior state-of-the-art methods such as StreetGaussians. Ablation studies further verify the importance of the instance-semantic loss and control points.

**Strengths:**

1. The method is technically sound and well-motivated.
2. The experiments and ablations are sufficient to support the main claims.

**Weaknesses:**

1. The distributed training setup for 3DGS is difficult to view as a novel contribution.
2. The paper does not compare against more recent SOTA methods such as NeuralRAD or SplatAD.
3. The generalization to unseen large-scale views (e.g., new lanes or unseen driving perspectives) is unclear.

**Questions:**

Please consider discussing the problems raised during weaknesses section.

---

> ### Author Response · Authors · 2025-11-16
>
> We would like to thank you for your thorough review and the valuable time you have devoted. In the following, we address your comments and concerns point by point. **All corresponding revisions have been incorporated into the manuscript and are highlighted in red for your convenience.**\
> \
> 1.**Novelty**. The core of our method does not lie in distributed training, so it is not listed separately as a contribution. Distributed training is included only to allow more researchers to perform dynamic scene reconstruction when single-GPU memory is limited. Unlike existing semantic methods, **the core of our approach** lies in jointly optimizing instance attributes using 2D instance ID masks and instance-segmented RGB images. This allows fast differentiation and control of Gaussian instances **without MLPs**. Meanwhile, we **disentangle scene and motion by enforcing instance motion consistency, spatial continuity, and local rigidity**, and employ **two instance-level losses** to optimize both the spatial distribution of Gaussians and the feature consistency. Compared to standard rasterization, we additionally incorporate both an instance rendering pipeline and an opacity rendering pipeline, and **reformulate the gradient computation for opacity.**\
> \
> 2.**We have added comparisons with the latest methods in Table 1 (corresponding to Table 2 in the paper)**.  NeuralRAD and SplatAD are excellent works, but both are constrained by a rigid-motion assumption and cannot directly edit different objects. Therefore, based on multiple reviewers’ suggestions, we chose OmniRe[1]—a decomposition method that is closer to our setting and also relies on the rigid assumption—as a primary comparison. In addition, we included the dynamic decomposition method Desire-GS[2] and the static method Octree-GS[3] to provide a more comprehensive comparison. The results show that our method achieves strong performance without requiring extra inputs or depth supervision, while enabling finer instance-level control. To better clarify our contributions, we refined the descriptions in the abstract and introduction. For the revised paper, we added **comparisons with various SOTA methods** in Table 2, incorporated **robustness analysis** in Figure 7, and included **segmentation performance analysis** and **a comparison of the Gaussian identification cost**  in rebuttal.\
> \
> **Table 1: Quantitative comparison of scene reconstruction results on the Waymo dataset. D$_{\text{opt}}$denotes depth optimization. “Box” indicates methods that utilize bounding boxes for dynamic modeling.SR and NVS indicate Scene Reconstruction and Novel View Synthesis, respectively.**
>
> | Methods   | Venue     | Box  | D$_{\text{opt}}$ | PSNR↑ (SR) | SSIM↑ (SR) | LPIPS↓ (SR) | PSNR↑ (NVS) | SSIM↑ (NVS) | LPIPS↓ (NVS) |
> | --------- | --------- | ---- | ---------------- | ------------- | ------------- | -------------- | ----------- | ----------- | ------------ |
> | 4D-GS     | CVPR2024  | ✗    | ✗                | 31.01         | 0.890         | 0.205          | 28.16       | 0.870       | 0.219        |
> | Deform-GS | CVPR2024  | ✗    | ✗                | 32.66         | 0.912         | 0.158          | 30.19       | 0.894       | 0.167        |
> | Octree-GS | TPAMI2025 | ✗    | ✗                | 33.63         | 0.936         | 0.103          | 32.13       | **0.927**   | 0.107        |
> | Desire-GS | CVPR2025  | ✓    | ✓                | 34.55         | 0.926         | 0.108          | 32.56       | 0.919       | 0.119        |
> | St-GS     | ECCV2024  | ✓    | ✗                | 35.66         | 0.936         | 0.109          | 30.87       | 0.882       | 0.139        |
> | S3-GS     | ICCV2025  | ✗    | ✓                | 33.44         | 0.927         | 0.118          | 31.62       | 0.917       | 0.124        |
> | OmniRe    | ICLR2025  | ✓    | ✓                | 34.95         | **0.943**     | 0.114          | 31.79       | 0.905       | 0.125        |
> | **Ours**  | N/A       | ✗    | ✗                | **36.12**     | 0.942         | **0.075**      | **34.49**   | **0.927**   | **0.082**    |

---

> ### Author Response · Authors · 2025-11-19
>
> 3.**Generalization Ability**. We illustrate the generalization ability of our model from two perspectives: novel-view synthesis within a single scene and adaptability across different scenes. **For a single scene**, since 3D Gaussians are optimized per scene, we evaluate generalization by synthesizing novel views. The novel view synthesis results in Tables 1 and 2 of the paper indicate that our method achieves excellent performance on unseen viewpoints, thanks to the inherent consistency of instance features and two instance-level losses that optimize the spatial distribution of Gaussians. **Across different scenes**, we further include the NuPlan dataset to demonstrate the generalization capability of the proposed approach, with the results presented in Table 2 (corresponding to Table 6 in the revised PDF). **Moreover**, compared to baseline methods, our approach does not rely on additional inputs and can better handle unseen object categories. Specifically, our method can explicitly model any object detected by the SAM, whereas box-based methods are limited to controlling only fixed categories with reliable 3D tracking. In addition, when the scene is expanded, transformed, or edited, our method can preserve the original instance results simply by increasing the feature dimensions, **without the need for retraining**.\
> **Table 2: Quantitative comparison of scene reconstruction results on the NuPlan dataset.**
> | Methods    | PSNR↑      | SSIM↑     | LPIPS ↓    |
> | ------------- | --------- | --------- | --------- |
> | Deform-GS [4]       | 28.10     | 0.880     | 0.143     |
> | HUGS [5]             | 24.16     | 0.768     | 0.274     |
> | Street Gaussians [6] | 26.05     | 0.813     | 0.180     |
> | OmniRe [1]           | 26.72     | 0.845     | **0.165** |
> | Ours             | **29.23** | **0.872** | 0.167     |
>
> [1] Ziyu Chen, Jiawei Yang, Jiahui Huang, Riccardo de Lutio, Janick Martinez Esturo, Boris Ivanovic, Or Litany, Zan Gojcic, Sanja Fidler, Marco Pavone, et al. Omnire: Omni urban scene reconstruction. arXiv preprint arXiv:2408.16760, 2024\
> [2] Chensheng Peng, Chengwei Zhang, Yixiao Wang, Chenfeng Xu, Yichen Xie, Wenzhao Zheng, Kurt Keutzer, Masayoshi Tomizuka, and Wei Zhan. Desire-gs: 4d street gaussians for static-dynamic decomposition and surface reconstruction for urban driving scenes. In Proceedings of the Computer Vision and Pattern Recognition Conference, pp. 6782–6791, 2025\
> [3] Kerui Ren, Lihan Jiang, Tao Lu, Mulin Yu, Linning Xu, Zhangkai Ni, and Bo Dai. Octree- gs: Towards consistent real-time rendering with lod-structured 3d gaussians. arXiv preprint arXiv:2403.17898, 2024.\
> [4] Ziyi Yang, Xinyu Gao, Wen Zhou, Shaohui Jiao, Yuqing Zhang, and Xiaogang Jin. Deformable 3d gaussians for highfidelity monocular dynamic scene reconstruction. In CVPR, 2024. 6, 7, 8, 12\
> [5] Hongyu Zhou, Jiahao Shao, Lu Xu, Dongfeng Bai, Weichao Qiu, Bingbing Liu, Yue Wang, Andreas Geiger, and Yiyi
> Liao. Hugs: Holistic urban 3d scene understanding via gaussian splatting. In CVPR, 2024. 2, 6, 7, 12\
> [6] Yunzhi Yan, Haotong Lin, Chenxu Zhou, Weijie Wang, Haiyang Sun, Kun Zhan, Xianpeng Lang, Xiaowei Zhou,
> and Sida Peng. Street gaussians: Modeling dynamic urban scenes with gaussian splatting. In ECCV, 2024. 2, 4, 5, 6, 7, 8, 12, 13

---

> ### Author Response · Authors · 2025-11-25
>
> To better highlight the contributions of this work, **we further clarify the differences from existing methods**.
>
> \
> (1) **Differences from instance-based methods.** Instance-based approaches such as **Gaussian-Grouping** [7] assign a high-dimensional vector to each Gaussian and classify it with an MLP. In contrast, our method substantially reduces both memory usage and runtime while achieving comparable segmentation quality, **as shown in Table 3**.  Methods such as  **OpenGaussian**[8] map high-dimensional instance encodings and spatial features to a finite set of prototype vectors. By contrast, our approach supports **plug-and-play** category expansion simply by increasing the feature dimensionality, without any retraining. Furthermore, feature-space consistency is achieved solely through the combination of instance RGB maps and an instance semantic loss, eliminating the need for a complex two-level codebook design.
>
> \
> **Table3: Results of Instance Distinction Method Comparison**.
> | Methods  | time(s) ↓     | Memory(Mb) ↓    | mIoU ↑       |
> |--------------------|----------------|----------------|--------------|
> | Gaussian Grouping [7]  | 0.2135         | 290.31         | 74.2         |
> | Ours                | 0.0051         | 259.79         | 71.7         |
> | Δ               | **-97.61%**    | **-10.51%**    | **-3.36%**   |
>
> \
> (2) **Differences from bounding-box–based methods**. Our method differs from bounding-box–based approaches such as OmniRe [1] and StreetGaussian [6] in four aspects. **a) Data cost**. Our method only relies on model-generated pseudo-labels to achieve high-quality instance-level disentanglement of scene and motion. In contrast, Box methods require more expensive human-annotated 3D information. **b) Motion Complexity.** Bounding-box–based methods typically assume that objects undergo rigid motion. For example, StreetGaussian explicitly states that it cannot handle non-rigid dynamic objects such as walking pedestrians. While this assumption captures local motion patterns, real-world motion is far more complex than a single rigid pattern. Our method inherits the idea of local rigidity and extends it by using multiple control points to jointly drive the motion of each entity, enabling a wider range of motion patterns. If additional semantic information is available, the sampling rate of control points can be adjusted to model motions with different levels of complexity. **c) Unknown categories**. Our method can explicitly model any object detected by the Segment Anything Model (SAM)  in the scene, whereas box-based approaches can only control fixed categories that have reliable 3D tracking. Moreover, our instance-level control operates at the Gaussian level, while box-based approaches operate at the box level, giving us much more flexible instance boundaries. **d) The inherent multi-view consistency of instance features** not only facilitates downstream applications but also benefits the optimization of Gaussians themselves.
>
> \
> **We sincerely appreciate your valuable feedback and would be glad to further discuss any concerns. We hope that our clarifications adequately address your questions**.
>
> \
> [7] Mingqiao Ye, Martin Danelljan, Fisher Yu, and Lei Ke. Gaussian grouping: Segment and edit anything in 3d scenes. In European Conference on Computer Vision, pp. 162–179. Springer, 2024.\
> [8] Yanmin Wu, Jiarui Meng, Haijie Li, Chenming Wu, Yahao Shi, Xinhua Cheng, Chen Zhao, Haocheng Feng, Errui Ding, Jingdong Wang, and Jian Zhang. Opengaussian: Towards point-level 3d gaussian-based open vocabulary understanding. In Proceedings of the Advances in Neural Information Processing Systems (NeurIPS), pp. 19114–19138, 2024b.

---

### Official Review · Reviewer_RntX · 2025-11-05

**Soundness:** 3
**Presentation:** 3
**Contribution:** 3
**Rating:** 4
**Confidence:** 4

**Summary:**

The paper proposes DDI‑Gaussian (DDIG), a dynamic 3D Gaussian splatting framework that targets instance‑level understanding and motion modeling in urban driving scenes. The core ideas are:

- assign each Gaussian a compact multi‑hot instance code to enable direct, per‑Gaussian instance discrimination;

- model motion using instance‑level sparse control points whose time‑conditioned offsets are predicted by a lightweight MLP and interpolated to dense Gaussians via KNN/LBS;

- introduce an instance‑level region loss (paired with an opacity map rendering pipeline) to suppress ghosting, and an instance‑semantic consistency loss;

- implement training/rendering in a distributed manner and supervise instance masks with SAM‑generated 2D pseudo‑labels fused across views.

**Strengths:**

- The paper argues convincingly that instance‑level modeling matters for urban dynamic scenes and that ghosting along motion trajectories is a persistent issue

- The multi‑hot per‑Gaussian instance code aims to avoid an auxiliary ID‑prediction MLP, and the instance‑level control‑point motion model reduces per‑Gaussian MLP inference, aligning with efficiency goals

- The paper details hybrid parallelism and dynamic load balancing leveraging Grendel‑style ideas. This is an interesting engineering incorporation.

**Weaknesses:**

- The paper says the per‑Gaussian instance embedding is “activated by a Sigmoid and then discretized via element‑wise rounding” to form a multi‑hot code. Rounding is non‑differentiable; the paper does not state whether a straight‑through estimator (STE) or an alternative surrogate is used so that BCE supervision on the rendered mask can back‑propagate into the instance codes. As written, gradients to the rounded codes would be zero almost everywhere, calling into question how the instance embeddings are actually learned. This is a critical technical omission.

- the text states “For all metrics, our method achieves the best performance.” But Table 1 (D32) shows StreetGaussian outperforming DDIG on PSNR (27.78 vs. 27.12)* and *SSIM (0.818 vs. 0.811)**—the very metrics focused on dynamic objects.

- While the paper displays instance maps and “semantic images” (e.g., Figure 4–5, p.8; Figure 9, p.14), there are no segmentation metrics (e.g., PQ, AP/AR, IoU) against any ground truth or even pseudo‑label references. Given that instance awareness is a primary contribution, the absence of standard instance/semantic quality metrics undermines the central claim.

- the paper defines view‑independent semantic features f_i and instance features e_i. But the instance‑semantic loss (Eq. 4, p.6) suddenly uses e to denote semantic features, reusing the same symbol previously assigned to instance features

- The paper positions itself against Gaussian Grouping and control‑point‑based deformation (SC‑GS, Superpoint‑GS) but does not provide direct quantitative comparisons to these methods

- The pipeline depends on SAM masks and a cross‑view association strategy “following Gaussian Grouping” (briefly mentioned in Sec 3.2, p.4), but there is no study of failure modes (mask fragmentation, ID switching) or sensitivity to SAM prompts/parameters. Since labels drive the instance code learning, robustness needs to be quantified.

- complete wall‑clock times, memory usage, Gaussian counts per method, and distributed scaling curves are missing or only anecdotally described

**Questions:**

- How are gradients propagated through the rounded instance codes?

- You mention following Gaussian Grouping to associate masks. What concrete association algorithm do you use (cost matrix, IoU thresholds, tracking heuristic), and how do you prevent ID collisions given the fixed 8‑bit multi‑hot code?

- Please clarify the symbols and dimensions

- Ablations vs. prior art.

- What value of \lambda is used in Eq. (5, p.6), and how sensitive is performance to it? Also, does the region loss ever delete small, fast‑moving objects (false negatives) due to mask misalignment from SAM?

---

> ### Author Response · Authors · 2025-11-16
>
> We would like to thank you for your thorough review and the valuable time you have devoted. In the following, we address your comments and concerns point by point. **All corresponding revisions have been incorporated into the manuscript and are highlighted in red for your convenience.**\
> \
> 1.**Mask association and ID conflicts**. Following Gaussian Grouping (GS-GP), we employ a well-trained zero-shot tracker DEVA[1] to propagate and associate masks, which ensures instance uniqueness. However, when the number of instances exceeds the predefined limit, GS-GP fails to operate reliably. **In contrast**, our method only requires expanding the dimensionality of the instance features to accommodate more instances, achieving several-fold scalability without retraining the Gaussians that have already been classified. **Furthermore**, ID duplication has a negligible impact on ghost removal, as our per-instance rendering strategy via instance selection confines the render area, avoiding artifacts at future timesteps. ID duplication is analogous to independently rendering two different vehicles while constraining their projected locations in the 2D image.\
> \
> 2.**Non-differentiability**. Rounding is used only when generating masks to select Gaussians, while the instance features are continuous values in [0,1]. During training, we compute the loss between the rendered 2D logits of the continuous instance features and the pseudo instance IDs, thereby making​ **the entire pipeline fully differentiable**. We have clarified this in the paper to avoid misunderstanding.\
> \
> 3.**Ablations vs. prior art**. We have added comparisons with the latest methods in Table 1 (corresponding to Table 2 in the paper), and includedl baselines in the ablation study for clearer comparison. The results show that our method achieves strong performance without requiring extra inputs or depth supervision, while enabling finer instance-level control.\
> **Table 1: Quantitative comparison of scene reconstruction results on the Waymo dataset.  D$_{\text{opt}}$ denotes depth optimization. “Box” indicates methods that utilize bounding boxes for dynamic modeling.SR and NVS indicate Scene Reconstruction and Novel View Synthesis, respectively.**
> | Methods   | Venue     | Box  |  D$_{\text{opt}}$| PSNR（SR）↑  | SSIM（SR)↑  | LPIPS(SR)↓  | PSNR↑ (NVS) | SSIM↑ (NVS) | LPIPS↓ (NVS) |
> | --------- | --------- | ---- | ----- | ------------- | ------------- | -------------- | ----------- | ----------- | ------------ |
> | 4D-GS     | CVPR2024  | ✗    | ✗     | 31.01         | 0.890         | 0.205          | 28.16       | 0.870       | 0.219        |
> | Deform-GS | CVPR2024  | ✗    | ✗     | 32.66         | 0.912         | 0.158          | 30.19       | 0.894       | 0.167        |
> | Octree-GS | TPAMI2025 | ✗    | ✗     | 33.63         | 0.936         | 0.103          | 32.13       | **0.927**   | 0.107        |
> | Desire-GS | CVPR2025  | ✓    | ✓     | 34.55         | 0.926         | 0.108          | 32.56       | 0.919       | 0.119        |
> | St-GS     | ECCV2024  | ✓    | ✗     | 35.66         | 0.936         | 0.109          | 30.87       | 0.882       | 0.139        |
> | S3-GS     | ICCV2025  | ✗    | ✓     | 33.44         | 0.927         | 0.118          | 31.62       | 0.917       | 0.124        |
> | OmniRe    | ICLR2025  | ✓    | ✓     | 34.95         | **0.943**     | 0.114          | 31.79       | 0.905       | 0.125        |
> | **Ours**  | N/A       | ✗    | ✗     | **36.12**     | 0.942         | **0.075**      | **34.49**   | **0.927**   | **0.082**    |
>
> \
> 4.**Robustness analysis and false negatives**. This is indeed important, and we have added the corresponding experiments in Section 4.4 (which corresponds to Figure 7 in the paper).  The results show that our method maintains stable multi-view performance even when instance IDs switch or their number is reduced. This is attributed to **the spatial constraints from the two instance-level losses, multi-view consistency of instance features, and joint determination of attribute offsets via control points. Figure 12 in the appendix further illustrates this.\
> \
> 5.**Instance quality metrics**.We have added the relal experiments, which are presented in the first paragraph of Appendix A.3. Specifically, **the proposed method achieves the panoptic quality (PQ), average precision (AP), and mean IoU (mIoU) of 62.1, 28.6, and 71.7**, respectively. Combined with experiments on scene reconstruction and instance-level decomposition, these results demonstrate that our approach can perform instance-level Gaussian distinction, effectively meeting the requirements of dynamic street scenes.

---

> ### Author Response · Authors · 2025-11-17
>
> 6.**Comparison with GS-GP and SC-GS**. We have added a comparison of the Gaussian identification cost with GS-GP in Table 2 . Experimental results show that our method achieves comparable segmentation quality while significantly reducing memory usage and runtime. In real-time dynamic scene construction, Gaussian Grouping’s per-frame Gaussian selection speed is insufficient to meet practical requirements.  For SC-GS, although our method adopts the idea of control points, their approach is restricted to a single dynamic object and cannot be directly applied to complex street scenes.\
> **Table2: Results of Instance Distinction Method Comparison**.
> | Methods  | time(s) ↓      | Memory(Mb) ↓    | mIoU ↑       |
> |--------------------|----------------|----------------|--------------|
> | Gaussian Grouping   | 0.2135         | 290.31         | 74.2         |
> | Ours                | 0.0051         | 259.79         | 71.7         |
> | **Δ**               | **-97.61%**    | **-10.51%**    | **-3.36%**   |\
>
> \
> 7.**Distributed setup**.  The main purpose of the distributed implementation is to allow researchers to reproduce our work even when single-GPU memory is limited. We did not conduct specific distributed training experiments because it is not the core focus of our study.  And our hardware environment lacked NVLink support. In our experiments, we use approximately 900K Gaussians, with training taking about 3 hours and 40 minutes.  Each GPU uses up to 21 GB of memory, and training is conducted on two GPUs in total.\
> \
> 8.**Hyperparameters**. In practice, λ is set to 10, and when λ takes values of 1, 10, 20, 50, 80, or 100, PSNR varies only within ±0.03.\
> \
> 9.**textbf Ambiguities and typos**. The PSNR errors and descriptive mistakes in Table 2 of the paper have been corrected. We thank you for noticing these carefully.\
> \
> 10. **PSNR\* and SSIM\***.  PSNR* and SSIM* are re-computed by projecting the 3D bounding boxes of dynamic objects onto the 2D image plane and calculating the pixel-wise loss only within the projected regions. **This evaluation metric is inherently more favorable to methods that rely on bounding boxes**. **Nevertheless**, as demonstrated by the visualizations and the the quantitative analysis, our method achieves more accurate scene reconstruction and a more reliable decomposition of dynamic objects.\
> \
> [1]Ho Kei Cheng, Seoung Wug Oh, Brian Price, Alexander Schwing, and Joon-Young Lee. Tracking anything with decoupled video segmentation. In Proceedings of the IEEE/CVF International Conference on Computer Vision, pp. 1316–1326, 2023.

---

> > ### Comment · Reviewer_RntX · 2025-11-24
> >
> > Thank you for your response. I have carefully reviewed the provided answers and the comments from fellow reviewers. However, I still have the following concerns:
> >
> > - Regarding mask association, how does the proposed method handle severe occlusion among dynamic objects? Specifically, are occluded objects assigned a new ID, or are they associated with their previous ID?
> >
> > - I noticed that the paper "3D STREETUNVEILER WITH SEMANTIC-AWARE 2DGS – A SIMPLE BASELINE" (ICLR 2025) was not cited or discussed. That work advocates using "hard labels" instead of "soft labels" to improve consistency. Could you clarify the advantages of using soft labels in your method compared to this approach?
> >
> > - Another relevant work, "OpenGaussian: Towards Point-Level 3D Gaussian-based Open Vocabulary Understanding" (NeurIPS 2025), also appears to have been overlooked. It introduces a two-level codebook learning mechanism for semantic association and claims similar benefits in terms of scalable instance support. Please clarify how your method differs or what advantages it offers over this approach.
> >
> > - In the response to Reviewer k2AD, it was mentioned that videos are available via an anonymous link. However, I was unable to access them. Could you please clarify this?

---

> ### Author Response · Authors · 2025-11-24
>
> **We sincerely thank the reviewer for carefully reading our response and providing valuable comments.** The StreetUnveiler and OpenGaussian works you mentioned are indeed excellent, and **we will cite them and add relevant discussions in the revised manuscript**. Below, we address your questions one by one.
>
> \
> 1.**Handling Severe Occlusions Between Dynamic Objects.** We handle severe occlusions among dynamic objects from two perspectives: pseudo-label generation and RGB-based semantic association. **(a) Robust Pseudo-labeling.** We leverage the XMem[1] architecture to maintain feature representations. Specifically, the global memory design provides inherent robustness against severe occlusion. During occlusion events, we do not assign a new ID to the target; instead, its tracklet is preserved. Upon re-emergence, the memory attention mechanism matches the target with its historical features to recover the original ID, ensuring long-term tracking consistency. A new ID is instantiated only when a strictly never-before-seen object is detected. **(b) RGB-Semantic Association.** We employ an RGB representation of instance labels to facilitate the optimization of instance IDs and Gaussian spatial positions. Specifically, we assign an additional instance color attribute (using only the DC component of Spherical Harmonics) to each Gaussian. We impose an instance-semantic loss to enforce ID consistency among Gaussians sharing the same RGB values, providing persistent and stable supervision even during occlusions. Moreover, instance IDs can be further associated based on feature similarity of their RGB values, ensuring global consistency. **As demonstrated in Figures 6, 7, and 12**, our method maintains global instance ID consistency across multiple frames, even in cases where pseudo-labels contain errors or omissions.
>
> \
> 2.**Advantages Over Hard Labels in StreetUnveiler**. In StreetUnveiler, hard labels are non-optimizable and rely on opacity and rotation attributes to render correct 2D semantic maps, primarily to simplify 2DGS removal. It presents several limitations compared to our soft label approach. **(a) Difficult to directly optimize and ignoring natural consistency of features**. Hard labels are non-differentiable and ignore the inherent multi-view consistency of features. In contrast, our soft label approach leverages this natural **multi-view consistency** during optimization. This significantly enhances geometric stability and robustness, effectively mitigating issues related to occlusion and optimization instability. **(b) Higher dependency on the semantic model**. Hard labels are prone to error propagation from the upstream semantic model. Our soft labels allow the model to converge to the correct category across multiple frames based on probabilistic statistics, thereby correcting initial segmentation errors. (c) **Boundary Quality**. In practice, a pixel represents a region rather than a single object, and hard labels can lead to boundary jaggies, unstable color optimization, and multi-view inconsistencies. Our proposed approach maintains the ability to directly segment objects through label discretization without requiring an MLP, while still benefiting from the advantages of soft labels.
>
> \
> 3.**Differences from OpenGaussian**. Our approach differs from OpenGaussian in terms of its objective, implementation strategy, and scalability. **(a) Mechanism for Identical Feature Consistency.** OpenGaussian employs a two-level codebook (combining instance and positional features) to ensure that spatially adjacent features are identical. In contrast, our approach eliminates the need for a codebook. We achieve identical feature consistency within the same semantic instance solely through our instance RGB maps combined with the instance semantic loss. **(b) Elimination of Codebook Dependencies.** OpenGaussian maps high-dimensional instance and spatial features to a finite set of prototype vectors. This introduces optimization instability, quantization errors, and limitations in handling an open-ended number of instance categories. **(c) Scalability.** Compared with OpenGaussian, our method supports category expansion in a plug-and-play manner—only a dimensionality increase is required, with no retraining involved. And, it allows direct control of different Gaussians via instance features without an additional MLP.
>
> \
> 4. We suspect that the link error is **due to a network issue**. I will further check it. **The specific video file can be viewed in the supplementary materials.** We apologize for the inconvenience caused.
>
> \
> **We thank you again for your valuable comments and look forward to any further discussion. We hope that our response has clarified your questions, and we would be grateful if you could consider raising your score accordingly.**
>
> \
> [1] Ho Kei Cheng and Alexander G Schwing. Xmem: Longterm video object segmentation with an atkinson-shiffrin memory model. In ECCV, 2022.

---

### Author Response · Authors · 2025-11-28

Dear Reviewers,

I hope this message finds you well. We remain eager to continue the discussion, despite the recent system issues. As the discussion period is coming to an end, **we sincerely hope that our responses have satisfactorily addressed all your concerns**. **If you have any further questions or feedback**, please feel free to let us know. Your insights are highly valuable and have greatly contributed to improving our work.

Thank you again for the time and effort you have devoted to reviewing our paper. **We wish you all the best in your future research and hope you will also meet reviewers as thoughtful and constructive as you have been to us**.

---

### Note · Authors · 2025-12-03

I have read and agree with the venue's withdrawal policy on behalf of myself and my co-authors.